# Theory-independent monitoring of the decoherence of a superconducting qubit with generalized contextuality

Albert Aloy [1,2] ✉, Matteo Fadel [3] ✉, Thomas D. Galley[1,2] ✉, Caroline L. Jones[1,2] ✉ & Markus P. Müller [1,2,4] ✉

Characterizing the nonclassicality of quantum systems under minimal assumptions is an important challenge for quantum foundations and technology. Here we introduce a theory-independent method of process tomography and perform it on a superconducting qubit. We demonstrate its decoherence without assuming quantum theory or trusting the devices by modelling the system as a general probabilistic theory. We show that the superconducting system is initially well-described as a quantum bit, but that its realized state space contracts over time, which in quantum terminology indicates its loss of coherence. The system is initially nonclassical in the sense of generalized contextuality: it does not admit of a hidden-variable model where statistically indistinguishable preparations are represented by identical hidden-variable distributions. In finite time, the system becomes noncontextual and hence loses its nonclassicality. Moreover, we demonstrate in a theory-independent way that the system undergoes non-Markovian evolution at late times. Our results extend theory-independent tomography to time-evolving systems, and show how important dynamical physical phenomena can be experimentally monitored without assuming quantum theory.

Demonstrating that some quantum systems have properties that genuinely defy classical explanation is at the forefront of current theoretical and experimental research. The strongest form of nonclassicality, Bell nonlocality[1], allows us to refute local hidden-variable models for experiments on spatially separated quantum systems. It is often experimentally more tractable to focus on single systems, avoiding the need for spacelike separation, and the Kochen-Specker theorem[2] precludes the existence of noncontextual hidden-variable models for such systems of Hilbert space dimension at least three.

The Kochen-Specker theorem relies, however, on several undesirable assumptions: it assumes the validity of quantum theory, the fact that the performed measurements are noiseless projective measurements, and that the outcomes depend deterministically on the underlying hidden variables. The latter two assumptions in particular are detrimental to the goal of the practical certification of nonclassicality. Spekkens' notion of generalized noncontextuality overcomes these difficulties[3]. In a nutshell, it stipulates that statistically indistinguishable operations (measurements, preparations or transformations) should have identical representations on the hidden-variable level. Generalized contextuality admits of robust experimental detection[4–6], and subsumes a variety of natural notions of nonclassicality (such as Wigner negativity[7,8]) into a single simple criterion.

Here, we certify the generalized contextuality of a superconducting qubit experimentally, directly from the measurement statistics and without assuming quantum theory. In contrast to earlier

[1]Institute for Quantum Optics and Quantum Information, Austrian Academy of Sciences, Vienna, Austria. [2]Vienna Center for Quantum Science and Technology (VCQ), Faculty of Physics, University of Vienna, Vienna, Austria. [3]Department of Physics, ETH Zürich, Zürich, Switzerland. [4]Perimeter Institute for Theoretical Physics, Waterloo, ON, Canada. ✉e-mail: Albert.Aloy@oeaw.ac.at; fadelm@phys.ethz.ch; Thomas.Galley@oeaw.ac.at; CarolineLouise.Jones@oeaw.ac.at; Markus.Mueller@oeaw.ac.at

experimental detection of Kochen-Specker contextuality in a super-conducting platform[9], our analysis thus makes significantly weaker assumptions. We analyze how the amount of contextuality changes over time—in particular, the system becomes noncontextual in finite time due to decoherence. In fact, we do more than this: we characterize the system completely, for different evolution times, by generalizing theory-agnostic tomography[10,11]. That is, by performing many preparation and measurement procedures, we determine its space of states and of effects (measurement outcomes) which, when combined linearly, reproduce the experimental statistics. Such pairs are known as generalized probabilistic theories (GPT)[12–15], with quantum theory and its qubit as a special case. For a quantum bit, the state space is the three-dimensional Bloch ball, but a GPT's state space can be any convex set of any dimension. Without assuming the validity of quantum theory, we determine directly from the data that our superconducting system's normalized state space is likely three-dimensional and in shape close to a ball. Observing the contraction of this "bumpy Bloch ball" yields a theory-independent monitoring of the quantum decoherence process, of its loss of contextuality, and of its non-Markovianity[16] at late times.

Superconducting qubits[17] admit the rapid implementation of a very large number of preparation and measurement procedures, which is necessary for the data-demanding sort of process tomography that we implement here. Our results hence demonstrate the suitability of the superconducting system for this sort of theory-independent analysis.

## Results

### Experimental setup

Our experiments are performed on a transmon superconducting qubit[18] with frequency $\omega_q = 2\pi \cdot 5.05$ GHz, hosted inside a three-dimensional readout cavity resonator with frequency $\omega_r = 2\pi \cdot 8.56$ GHz. Strong dispersive coupling between the two allows us to measure the qubit's state by performing a transmission measurement on the resonator to probe its qubit-state-dependent resonance frequency, as standard in circuit-QED[19]. This signal is amplified by both cryogenic and room temperature amplifiers, which give a readout fidelity of 85(1)%. We measure a qubit energy relaxation rate $T_1 = 21.9 \pm 0.4\,\mu$s and Ramsey decoherence rate $T_2^* = 12.7 \pm 0.6\,\mu$s, as well as a resonator frequency shift $\chi = -2\pi \cdot 1.89$ MHz when the qubit is excited. Preparation of the initial state is achieved by applying a 200 ns resonant Rabi pulse with well-defined amplitude and phase to the qubit $t = 0$ state $|0\rangle$. This amplitude and phase are chosen from a list $\mathcal{V}_m$

defining the space of $m$ states. In order to have a uniform distribution of $m$ points on the surface of the Bloch sphere, we chose a Fibonacci distribution[20]. We then wait for a variable time $\tau$, after which a second Rabi pulse is applied in order to specify the measurement basis. Again, amplitude and phase of the pulse are chosen from a list $\mathcal{V}_n$, now specifying the space of effects. In our case, $m = n = 100$, and the two lists of preparations and measurement directions coincide. Finally, the qubit's state is measured in the computational $z$-basis $\{|0\rangle, |1\rangle\}$ by the standard circuit-QED readout technique in the strong-dispersive-coupling regime, as mentioned above. For each combination of state preparation and measurement, this sequence is repeated 2000 times in order to average over the qubit projection noise. This gives us the frequencies $F_{ij}$ from which we estimate the probabilities $p(0|P_i, M_j)$ entering Eq. (3) below.

We describe the experimental setup and the qubit readout characterization procedure in more detail in the Methods' subsection "Experimental setup and qubit readout characterization".

### Theory-independent analysis

To provide a theory-independent analysis, we view the experimental setup as a prepare-and-measure scenario, see Fig. 1. We have $m$ possible preparation procedures $P_i$ and $n$ measurement procedures $M_j$, each with outcome $a \in \{0, 1\}$. Moreover, we have the option to introduce a waiting time $\tau$ between preparation and measurement. Without loss of generality, we choose the convention that this evolution for time $\tau$ is considered a part of the preparation procedure, yielding effective preparation procedures $P_i^\tau$. Thus, we probe the system (the superconducting qubit) to estimate the conditional probabilities $p(a|P_i^\tau, M_j)$. By normalization, it is sufficient to obtain the statistics for one of the outcomes which we denote by $a = 0$ – shown in Fig. 2 for $\tau = 0$. Statistics are gathered for all the possible $i, j$ and for $\tau = \{0, 5, 10, 15, 20, 30, 40, 50\}\,\mu$s. For every fixed value of $\tau$, this defines a corresponding *operational theory*[3]: a collection of preparation and measurement procedures together with a probability rule that describes the statistical properties of the laboratory system.

Following this operational approach, generalized noncontextuality[3,21] requires that procedures which are statistically equivalent at the operational level (i.e., producing identical statistics if all other processes are fixed) must also be identically represented in the underlying hidden-variable model. A hidden-variable model (also

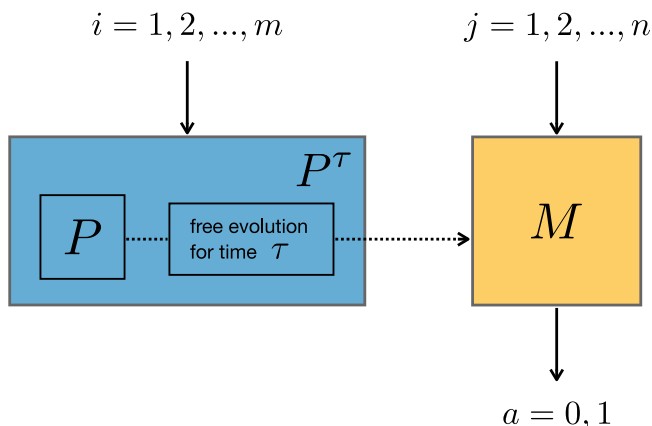

**Fig. 1 | Diagram of the prepare-transform-measure setup of the experiment.** We regard the actual state preparation and the subsequent time evolution as a single preparation procedure, which is simply a convenient convention (analogous to the choice of Schrödinger versus Heisenberg picture). As a result, the prepared states will depend on the waiting time $\tau$, but the measurements will not.

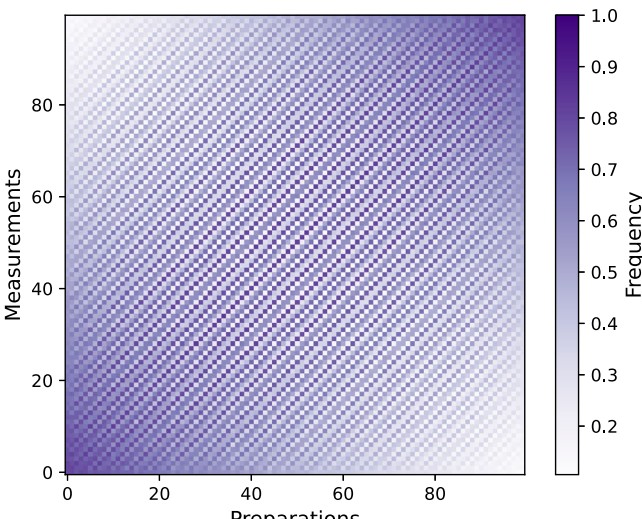

**Fig. 2 | Frequency data collected in the experiment.** Experimentally measured frequency of occurrence of the outcome $a = 0$ for the $m \times n$ table of preparations and measurements, for $\tau = 0$.

**Fig. 3 | Relations between the state spaces of the qubit, stabilizer qubit[31] and classical four-level system. a** The state space of the stabilizer qubit (in green) is embedded in the state space of the qubit, given by the gray ball; **b** the state space of the stabilizer qubit (in green) is embeddable within a tetrahedron (in gray), the state space of a classical four level system; **c** the embedding of the stabilizer qubit into the tetrahedron pictured here is such that any valid effect on the stabilizer qubit is also a valid effect on the tetrahedron. This requirement prevents, for example, the embeddability of the qubit into the tetrahedron (we know that the qubit is nonembeddable because it is contextual). Here the orange planes represent the effect corresponding to the +1 outcome of the Pauli $Z$ measurement: the upper plane intersects all states giving probability 1 for the outcome +1, the central plane those giving probability $\frac{1}{2}$ and the lower plane those giving probability 0.

known as an ontological model) of an operational theory is defined on a measurable space $\Lambda$. To each preparation $P$ in the theory, there corresponds a probability distribution $\mu_P(\lambda)$ on $\Lambda$, and to every outcome $a$ of every measurement $M$, there corresponds a response function $\chi_{a|M}(\lambda)$, such that $\sum_a \chi_{a|M}(\lambda) = 1$. The assignments are such that the probabilities are reproduced by the laws of classical probability theory,

$$p(a|P, M) = \int_\Lambda \chi_{a|M}(\lambda)\mu_P(\lambda)\mathrm{d}\lambda. \quad (1)$$

A hidden-variable model is preparation-noncontextual[3,21] if any two operationally equivalent preparations $P$ and $P'$ have $\mu_P(\lambda) = \mu_{P'}(\lambda)$. Two preparations $P$ and $P'$ are operationally equivalent if

$$p(a|P, M) = p(a|P', M) \text{ for all } a, M. \quad (2)$$

Similarly, the model is measurement-noncontextual if operationally equivalent measurement outcomes (i.e., ones that have identical probabilities for all preparation procedures) are represented by the same response function.

For our scenario, we may not only consider the actually implemented procedures $M_i$ and $P_j$, but also statistical mixtures of those, such as the preparation procedure $P_{\mathrm{mix}}$ which results in implementing either preparation $P_1$ or $P_2$ with probability $\lambda$ or $1 - \lambda$, respectively. We can then represent operationally equivalent preparation procedures $P$ and $P'$ by some *state* $\boldsymbol{s}_P = \boldsymbol{s}_{P'}$, an element of some vector space over the real numbers. Statistical mixtures are then represented by convex combinations, such that, for example, $\boldsymbol{s}_{P_{\mathrm{mix}}} = \lambda \boldsymbol{s}_{P_1} + (1 - \lambda)\boldsymbol{s}_{P_2}$. Similarly, we can represent operationally equivalent measurement-outcome pairs $(a, M)$ and $(a', M')$ as so-called *effects* $\boldsymbol{e}_{a,M} = \boldsymbol{e}_{a',M'}$, elements of the dual space, such that $p(a|P, M) = \langle \boldsymbol{e}_{a,M}, \boldsymbol{s}_P \rangle$. All possible states of a system form a convex set $\mathcal{S}$, and all possible effects yield another convex set $\mathcal{E}$. This defines a generalized probabilistic theory (GPT)[14,15], sometimes also called a "GPT system". Please see the "Methods" section for more details, and for how a quantum bit is described in this formalism. In particular, its space of pure states is the famous Bloch sphere, and the full space $\mathcal{S}$ of all (pure and mixed) states is the Bloch ball.

Quantum and classical systems are examples of GPT systems, but the framework contains many more exotic possibilities, such as theories with superstrong nonlocality[12] and higher-order interference[22,23]. Originally, interest in GPTs originated in the research program to reconstruct the abstract formalism of quantum theory from simple physical principles[24–27]. Here, however, we use the GPT formalism as a tool for a theory-independent description of our experimental statistics. In particular, knowing the GPT associated to an operational theory allows us to determine whether it admits of a preparation- and measurement-noncontextual hidden-variable model, using the criterion of *simplex embeddability* demonstrated in ref. 28. If it does admit of such a model, we say that the physical system is noncontextual. This can be checked with a linear program[29,30].

For example, it can be checked that the qubit GPT system is contextual, i.e. cannot be embedded into any classical probability simplex, in contrast to the stabilizer qubit, a GPT system defined by restricting the qubit to convex combinations of its stabilizer states and to stabilizer measurements[31], as illustrated in Fig. 3. Hence, the stabilizer qubit is noncontextual (when restricted to preparations and measurements) or "classical" in this sense. This is related to the Gottesman-Knill theorem, establishing the efficient classical simulatability of stabilizer quantum computation[32]. More generally, generalized contextuality is a resource for a number of information processing tasks[33–38].

The obstacle to applying this directly to our experiment is that we do not know the probabilities $p(a|P, M)$, but only experimentally determined approximations, namely the frequencies of occurrence of the outcomes (depicted in Fig. 2). Hence, we cannot directly write down the GPT that describes our superconducting system, but have to estimate the GPT from the experimental data. We do so by modifying and generalizing the method of theory-agnostic tomography[10,11], which leads to the following multi-step procedure.

Let us first discuss the procedure for a fixed value of waiting time $\tau$; say, $\tau = 0$. We start by collecting the experimental statistics for a large set of possible preparations $P_i$ and two-outcome measurements $M_j$, with $i \in \{1, ..., m\}$ and $j \in \{1, ..., n\}$. The goal is to estimate the conditional probabilities $p(0|P_i, M_j)$ for obtaining outcome $a = 0$, given preparation $P_i$ and measurement $M_j$. We organize the collection of statistics in the form of an $m \times n$ matrix $D$ as:

$$D = \begin{pmatrix} p(0|P_1, M_1) & p(0|P_1, M_2) & \cdots & p(0|P_1, M_n) \\ p(0|P_2, M_1) & p(0|P_2, M_2) & \cdots & p(0|P_2, M_n) \\ \vdots & \cdots & \ddots & \vdots \\ p(0|P_m, M_1) & p(0|P_m, M_2) & \cdots & p(0|P_m, M_n) \end{pmatrix}. \quad (3)$$

In an actual experiment, the conditional probabilities $p(0|P_i, M_j)$ are estimated by running the experiment $N$ times. Thus, when talking about experimental data, instead of the matrix $D$ we will refer to a matrix $F$ containing the observed frequencies $F_{ij} = f(0|P_i, M_j)$. i.e., $F$ is a frequency table. Given that there are $N$ runs for each pair of preparation and measurement $(P_i, M_j)$, we have $F_{ij} = \frac{N_{ij}}{N}$, where $N_{ij}$ is the number of outcomes 0 observed for the pair $(P_i, M_j)$. In the case where a large number $M$ of frequency tables $\{F^q\}_{q=1}^M$ are obtained, one can directly compute the variance of a frequency table $F^q$ as $(\Delta F_{ij}^q)^2 = (F_{ij}^q - \bar{F}_{ij})^2$, where $\bar{F}_{ij} = \frac{1}{M}\sum_{q=1}^M F_{ij}^q$. In the case where one does not have a large

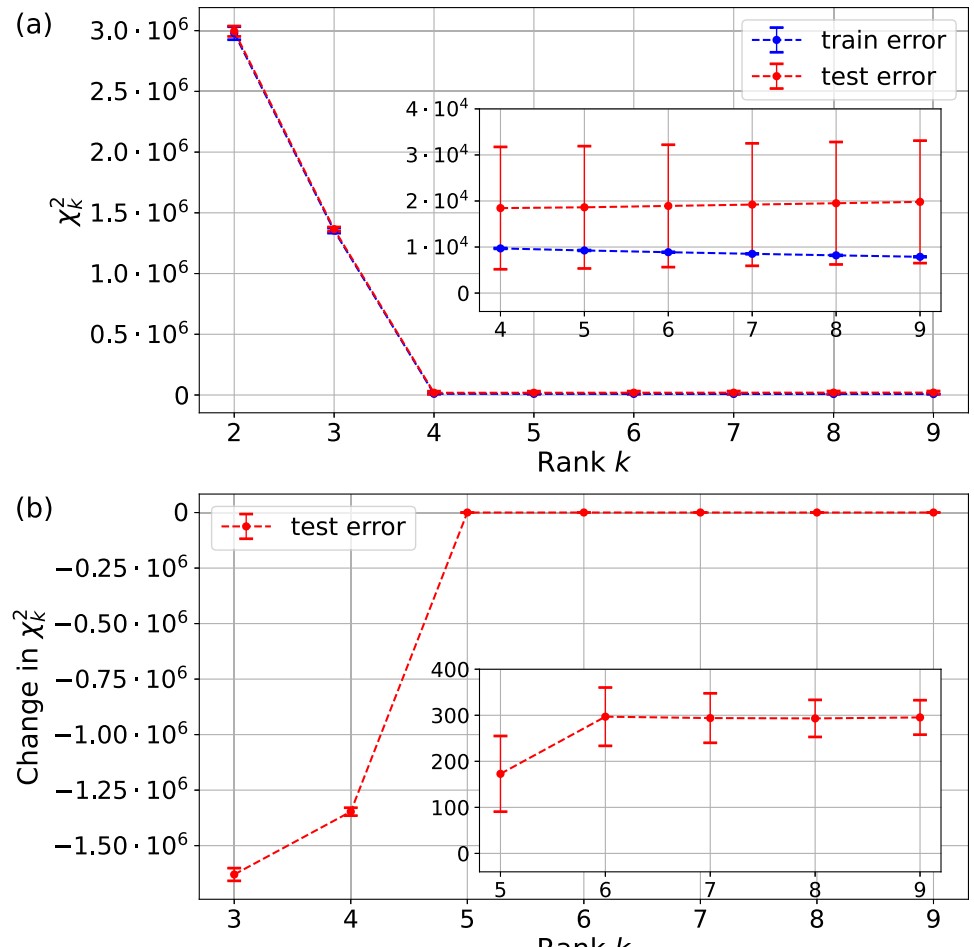

**Fig. 4 | Errors of the optimal GPT fits for different ranks. a** Test and train errors for the optimal GPT fits for ranks $k \in \{2, ..., 9\}$. Inset: zoomed in for ranks $k \geq 4$. The training error is evaluated on 10 frequency tables $\{F^\alpha\}_{\alpha=1}^{10}$, whilst the test error is evaluated on the 90 pairs of frequency tables $(F^\alpha, F^\beta)$ with $\alpha \neq \beta$. The error bars for the training error and test error are given by the statistical uncertainty over the 10 number of frequency tables, we can estimate $(\Delta F_{ij})^2$ by making the following assumptions[10]. We expect that in the limit of $N \to \infty$, these frequencies converge to the conditional probabilities $p(0|P_i, M_j)$, under the assumption of ideal repeatability disregarding drift and other imperfections. In this case, the variable $N_{ij}$ is the number of 0 outcomes of $N$ independent events in a sequence of experiments. This can be modeled as a binomial distribution with $N$ events and probability given by the frequency $F_{ij} = \frac{N_{ij}}{N}$. The variance in $N_{ij}$ is therefore $\frac{N_{ij}(N-N_{ij})}{N}$, which implies that the variance in the frequency is therefore $(\Delta F_{ij})^2 = \frac{N_{ij}(N-N_{ij})}{N^3} = \frac{F_{ij}(N-N_{ij})}{N^2} = \frac{F_{ij}(1-F_{ij})}{N}$.

The matrix $D$ contains all relevant statistical information about the prepare-and-measure experiment. For every GPT, the probabilities $p(0|P_i, M_j)$ are given by $\langle e_j, s_i \rangle$, where the effects $e_j$ describe the measurement and its outcome, and the states $s_i$ describe the preparation procedures. States are elements of some real vector space of unspecified dimension $k$ and effects elements of the dual space, with $\langle , \rangle$ denoting the natural pairing, i.e., the application of the covector $e_j$ to the vector $s_i$, yielding a real number. The sets of all possible states and effects define the GPT and determine its information-theoretic and physical behavior. Our goal now is to find the GPT state and effect spaces that best fit for $D$ while minimizing the number of parameters. That is, we want to find a fitting GPT model which assigns a $k$-dimensional state vector $s_i$ to each preparation $P_i$ and a $k$-dimensional effect

tables and 90 pairs, respectively. **b** For each pair $(F^\alpha, F^\beta)$ with $\alpha, \beta \in \{1, ..., 10\}$, $\alpha \neq \beta$ with $F^\alpha$ serving as the training data and $F^\beta$ as the testing data we plot $\chi_k^2(F^\beta, D_k^\alpha) - \chi_{k-1}^2(F^\beta, D_{k-1}^\alpha)$ which is the change in the test error between the best fit rank $k$ and best fit rank $k-1$ models. For $k \leq 4$ it is strictly negative, showing that the test error decreases, while for $k > 4$ it is strictly positive, showing that the test error increases.

vector $e_j$ to each measurement outcome 0 of $M_j$. Thus, the $m \times k$ matrix $S = (s_1, s_2, ..., s_m)^\top$ and the $k \times n$ matrix $E = (e_1, e_2, ..., e_n)$ can be used to factorize the data table $D = SE$, resulting in an $m \times n$ matrix of rank $k$, this time with entries $D_{ij} = s_i^\top e_j$ corresponding to the probabilities $p(0|P_i, M_j)$ predicted by the GPT.

To perform theory-agnostic tomography in our experiment, we first have to determine the appropriate rank $k$. To this end, we fit a rank $k$ matrix $D = SE$ to the experimentally obtained frequency table $F$ for various values of $k$. The best rank $k$ matrix $D$ is determined via minimization of the weighted $\chi_k^2$ in the following optimization problem[10]:

$$\min_{D \in M_{mn}} \quad \chi_k^2 = \sum_{ij} \frac{(F_{ij} - D_{ij})^2}{\Delta F_{ij}^2} \tag{4}$$
$$\text{subject to} \quad \text{rank}(D) = k, \ 0 \leq D_{ij} \leq 1.$$

In the Methods' subsection "Solving the weighted low-rank approximation problem of Eq. (4)", we describe how we solve this nonconvex optimization problem numerically, following the method presented in ref. 10.

The experimental data consists of ten frequency tables $\{F^\alpha\}_{\alpha=1}^{10}$ for the 100 preparations and measurements specified in section "Experimental setup". For every $k \in \{2, ..., 9\}$, the average optimal $\chi_k^2$ over $F^\alpha$ is plotted in blue in Fig. 4a. It decreases sharply for $k < 4$ and decreases slowly for $k \geq 4$. The large $\chi_k^2$ values for $k < 4$ indicate that the models $D_k$

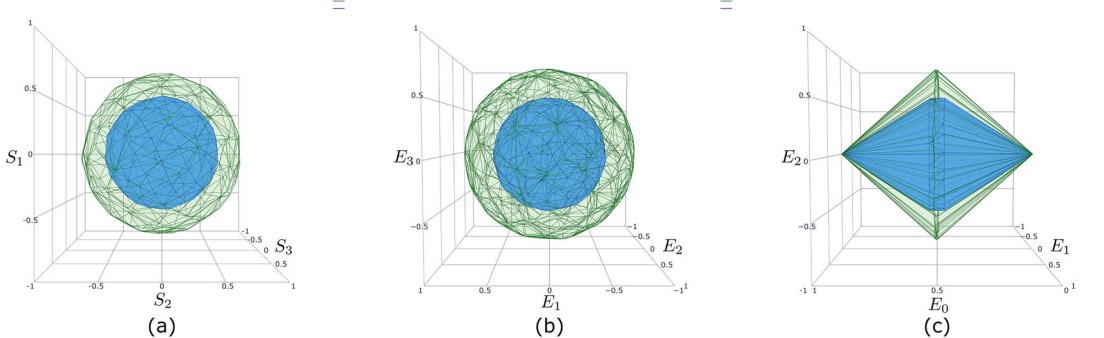

**Fig. 5 | Reconstructed state and effect spaces for zero time delay.** Plots showing the realized (blue) and consistent (green) spaces for the rank $k = 4$ GPT for $\tau = 0$. **a** The normalized state spaces are three-dimensional and similar to Bloch balls; **b** effect space projection onto dimensions 1,2,3; **c** effect space projection onto dimensions 0,1,2 (or similarly for 0,2,3 and 0,1,3) containing the zero and unit effects 0 and **1**.

have too few parameters to properly account for the regularities in the data; they underfit the data. For $k > 4$, the $\chi_k^2$ decrease at a slower rate as the extra parameters in the models fit to the statistical fluctuations in the frequency tables. The inflection point at $k = 4$ indicates that this is likely the rank of the underlying 'true' GPT generating the probability table $D$ of Eq. (3).

Fitting to statistical fluctuations is a signature of overfitting of a model. Following ref. [11], we can detect overfitting of a model by evaluating the optimal rank $k$ fit $D_k$ obtained from a frequency table $F^\alpha$ on a different frequency table $F^\beta$. $F^\alpha$ is known as the *training data* (since it is used to train the model $D_k$) and $F^\beta$ is known as the *test data* (since it is used to evaluate the model $D_k$). Since the statistical fluctuations of $F^\alpha$ and $F^\beta$ are independent, the more a model $D_k$ fits the statistical fluctuations in $F^\alpha$, the worse it will perform when evaluated on $F^\beta$. Given a training data/test data pair $(F, F')$ the test error relative to a model $D$ is

$$\chi_k^{2\,\text{test}} = \sum_{ij} \frac{(F'_{ij} - D_{ij})^2}{(\Delta F'_{ij})^2}. \qquad (5)$$

The test error is expected to be high for models which underfit the data (since they have too few parameters to adequately capture the structure of the data), and expected to decrease as $k$ approaches the true underlying rank $k_{\text{true}}$. For $k > k_{\text{true}}$ the test error is expected to increase. As such, the test error is expected to be minimal for the value of $k$ which neither underfits nor overfits the data, which is expected to be $k = k_{\text{true}}$.

Figure 4a shows the average test error for each $k \in \{2, \ldots, 9\}$ evaluated for every pair $(F^\alpha, F^\beta)(\alpha \neq \beta)$, where $F^\alpha$ is the training data and $F_\beta$ is the test data. The large statistical variances in the test error in Fig. 4a do not allow us to infer that the test error reaches its minimum at rank 4. In Fig. 4b, we plot the average difference in test error $\chi_k^2(F^\beta, D_k^\alpha) - \chi_{k-1}^2(F^\beta, D_{k-1}^\alpha)$. The error bars are sufficiently small to allow us to conclude that the change in test error is negative for $k \leq 4$ and positive for $k > 4$, allowing us to conclude that the minimal test error occurs for $k = 4$, and thus that the rank of the underlying true GPT is likely $k_{\text{true}} = 4$.

## Dynamical analysis: decoherence

To determine the dimension of the GPT system, we restrict our attention to the $\tau = 0$ data. In Fig. 4a, we have computed the optimal models fitting the experimental data $F$ for different rank $k$ candidates with $k \in \{2, \ldots, 9\}$. In particular, following Eq. (4), we minimize $\chi_k^2$ for each candidate $k$ obtaining the best-fit GPT models of rank $k$ to the obtained experimental data.

We have seen that solving Eq. (4) provides a model with the best-fit of rank $k$ to the probability matrix $D$ sampling the observed measurement outcomes. Moreover, we have observed that rank $k = 4$ provides the best estimate of an underlying model describing the

experiment. This could correspond to a qubit description as expected: the dimension of the set of normalized states will then be $k - 1 = 3$, which coincides with the dimension of a qubit's Bloch ball state space.

Let us now characterize the state and effect spaces for the GPT model of rank $k = 4$, which generate $D$. To obtain $D$, we have split the problem into estimating the realized GPT states $\mathcal{S}$ and effects $\mathcal{E}$. The decomposition $D = SE$ is not unique: for every invertible $k \times k$-matrix $L$, we also have $D = (SL)(L^{-1}E)$, and all possible decompositions are of this form, see Methods' subsection "Uniqueness of the decomposition $D = SE$". Following ref. [10], we choose a parametrization where the first component of a state denotes its normalization, and so the first column of $S$ consists entirely of ones. Furthermore, the first column of $E$ is chosen to be the unit (normalization) effect. All further freedom in choosing $L$ only leads to a different linear reparametrization of the resulting GPT, and this does not change any of its physical properties (see the notion of "equivalent GPTs" in Methods' subsection "Generalized probabilistic theories".

Figure 5a shows the resulting state space in blue, and Fig. 5b, c shows projections of the effect space in blue for one specific parametrization (see Methods' subsection "Generalized probabilistic theories"). As expected, $\mathcal{S}$ resembles a quantum bit Bloch ball, and $\mathcal{E}$ resembles the set of Hermitian $2 \times 2$ operators $E$ with eigenvalues in [0, 1] (which is $(k = 4)$-dimensional). However, in contrast to a perfect quantum bit, $\mathcal{S}$ and $\mathcal{E}$ are polytope and not exact duals of each other since only finitely many preparations and measurements have been implemented. That is, given our realized set of effects $\mathcal{E}$, we can consider the set $\mathcal{S}_{\text{consistent}}$ of all possible vectors $s$ which give valid probabilities for all measurements, i.e., $0 \leq \langle e, s \rangle \leq 1$ for all effects $e \in \mathcal{E}$. Then $\mathcal{S} \subsetneq \mathcal{S}_{\text{consistent}}$. Similarly, given $\mathcal{S}$, we can consider the set $\mathcal{E}_{\text{consistent}}$ of all possible covectors $e$ that give valid probabilities on all states, and $\mathcal{E} \subsetneq \mathcal{E}_{\text{consistent}}$. These sets are shown in green in Fig. 5.

For the sake of the argument, assume for a moment that the no-restriction hypothesis holds[39], as predicted, e.g., by quantum theory: all possible outcome probability rules are implementable effects, and all possible vectors yielding valid probabilities on the possible measurements are implementable states. In the hypothetical (but unrealistic) case that the measurements have been perfectly noiseless and all possible measurements have actually been implemented, this would imply that $\mathcal{S}_{\text{consistent}}$ describes the set of all possible states of the system, and $\mathcal{S}$ being a strict subset of this means that the preparations have been inherently noisy. Realistically, both preparations and measurements are noisy, but a theory-independent analysis cannot tell us whether the error occurred on the level of the former or the latter. All our analysis can tell us is that the no-restriction hypothesis is violated by the experimentally realized GPT system, and the considerations below allow us to quantify how strong the violation is.

Consider the distinguishability of two states $s$ and $s'$, defined as $\mathcal{D}(s, s') := \max_{e \in \mathcal{E}} |\langle e, s \rangle - \langle e, s' \rangle|$. For ideal quantum systems, this equals

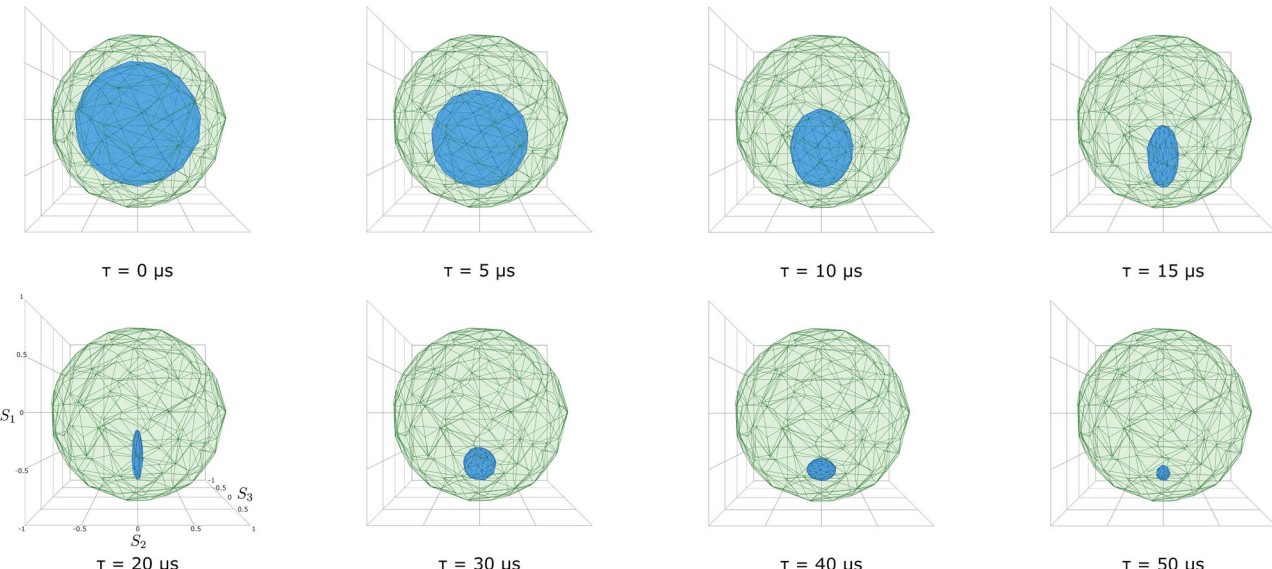

**Fig. 6 | Evolution of the state space for different delay times.** Plots of the consistent state space $\mathcal{S}_{\text{consistent}}$ (green) for all preparations, and realized state spaces $\mathcal{S}^\tau$ (blue) for varying wait times $\tau = [0, 5, 10, 15, 20, 30, 40, 50]$ μs.

the trace distance between the two density matrices that represent the states, $\frac{1}{2} \| \rho_s - \rho_{s'} \|_1$. For any given state $s$, consider the function $f(s) := \max_{s' \in \mathcal{S}} \mathcal{D}(s, s')$, then $0 \le f(s) \le 1$, and this equals 1 if and only if $s$ is perfectly distinguishable from some other state. As shown in ref. 40, the no-restriction hypothesis implies that $f(s) = 1$ for all boundary points $s$ of the state space, the "perfect distinguishability" axiom of ref. 40. In our case, this function takes values in between $0.681 \pm 0.003$ and $0.691 \pm 0.004$, whose minimum on the boundary points gives us a theory-independent quantity saying "how restricted" the effective GPT system is. Under the additional assumption that quantum theory holds, and that the superconducting system is fundamentally described by a qubit, this tells us that there must have been states $\rho_s$ prepared (i.e. $s \in \mathcal{S}$) that are close to pure, in the sense that their largest eigenvalue is $\lambda_{\max}(\rho_s) \ge 0.846 \pm 0.002$ (and, assuming approximate rotational symmetry as apparent in Fig. 5, all prepared states for $\tau = 0$ should have this property approximately); see Methods' subsection "Lower bound on the purity of the prepared states". Note that $\lambda_{\max}(\rho_s) = \langle \psi | \rho_s | \psi \rangle$, i.e., the fidelity between the prepared state $\rho_s$ and its eigenstate $|\psi\rangle$ corresponding to its largest eigenvalue.

Let us now turn to the task of monitoring the time evolution of our superconducting system, via snapshots for different evolution times $\tau$. We modify and extend the method of theory-agnostic tomography as follows. For a finite set of $\{\tau_0, \tau_1, \dots \tau_T\}$, we can obtain the frequency tables $\{F^\tau\}$. Since the measurements for each $F^\tau$ are treated to be the same by convention (recall Fig. 1), we can write a total frequency table $F$, which is a $m(T + 1) \times n$ matrix:

$$F = \begin{pmatrix} F^{\tau_0} \\ F^{\tau_1} \\ \vdots \\ F^{\tau_T} \end{pmatrix} \qquad (6)$$

The probability table for a rank $k$-GPT in this set of prepare-and-measure experiments with shared measurements is

$$D = \begin{pmatrix} D^{\tau_0} \\ D^{\tau_1} \\ \vdots \\ D^{\tau_T} \end{pmatrix} = \begin{pmatrix} S^{\tau_0} \\ S^{\tau_1} \\ \vdots \\ S^{\tau_T} \end{pmatrix} E = SE, \qquad (7)$$

with $S$ an $m(T + 1) \times k$ matrix and $E$ a $k \times n$ matrix.

The problem of finding the best rank-$k$ GPT fit for the sequence of prepare-and-measure scenarios $\{(\mathcal{P}^{\tau_0}, \mathcal{M}), (\mathcal{P}^{\tau_1}, \mathcal{M}), \dots, (\mathcal{P}^{\tau_T}, \mathcal{M})\}$ is thus equivalent to the task of finding the best GPT rank-$k$ GPT fit for $m(T + 1)$ preparations and $n$ measurements $\{\mathcal{P}', \mathcal{M}\}$, where $\mathcal{P}' = \bigcup_i \mathcal{P}^{\tau_i}$.

The result is shown in Fig. 6. From our knowledge of quantum physics and the experimental platform, we expect two effects to be in place: decoherence and relaxation to the ground state. Both effects are visible in the theory-independent representation. For every GPT, reversible transformations are represented by linear symmetries of the state space. Hence, under reversible time evolution, $\mathcal{S}$ is preserved. For $\tau \le 20$ μs, we see, however, that this is not the case, and that the time evolution is manifestly irreversible, shrinking $\mathcal{S}$ towards more mixed states. For all times, we see that evolution moves the state space closer to a single distinguished state (potentially a pure state), which describes a relaxation process. Quantum physics intuition tells us that this should correspond to the superconducting qubit's ground state, but our theory-independent analysis does not allow us to conclude this with certainty.

## Contextuality and its loss under decoherence

Now that we have a description of the different $\mathcal{S}^\tau$ and $\mathcal{E}^\tau$, we can check whether the corresponding GPT systems are noncontextual. To do so, we use the linear program of ref. 30 to determine whether the GPT systems are simplex-embeddable. Every system becomes noncontextual if a sufficient amount of noise is added. To quantify this, we use the state $\mu^\tau$, which is the average of all extremal states of $\mathcal{S}^\tau$ (think of it as the maximally mixed state in the center of $\mathcal{S}^\tau$), and we consider depolarizing noise that replaces every given state by $\mu^\tau$ with probability $r$. The resulting state space $\mathcal{S}^\tau_r := (1 - r)\mathcal{S}^\tau + r\mu^\tau$ will be noncontextual if $r$ is large enough. In Fig. 7, we show how large $r$ has to be chosen such that $\mathcal{S}^\tau_r$ is noncontextual. If the answer is $r > 0$, then the system at time $\tau$ is contextual, and otherwise, if $r = 0$, it is noncontextual. The uncertainties in Fig. 7 (resp. Fig. 8) are given by the standard deviations in the robustness (resp. relative volume) computed from 7 frequency tables $\{F^\alpha\}_{\alpha=1}^7$ of the form given in Eq. (6).

We see that the system is initially contextual, but that it looses its contextuality between the times of $\tau_2 = 10$ μs and $\tau_3 = 15$ μs. In other words, the decoherence process leads to the system becoming effectively classical and remaining so. Note that the error bars around $r = 0$ are zero, because small perturbations of noncontextual systems are

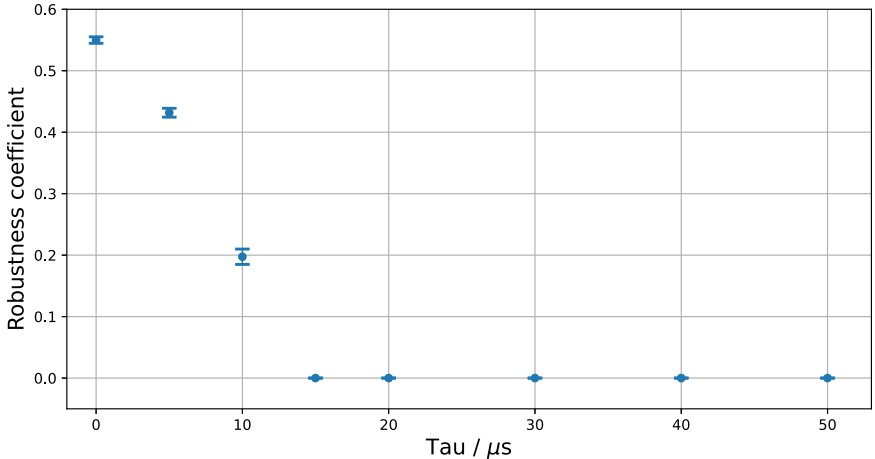

**Fig. 7 | Degree of contextuality of the GPT systems for different delay times.** Plot showing the robustness coefficient, i.e., the necessary amount of depolarizing noise $r$ that has to be added such that a noncontextual ontological model exists for the GPT system, tested for increasing delay time $\tau$ (with 7 runs per value of $\tau$, with the standard deviation then determining error bars). For robustness $r = 0$, the superconducting system is noncontextual, and otherwise contextual.

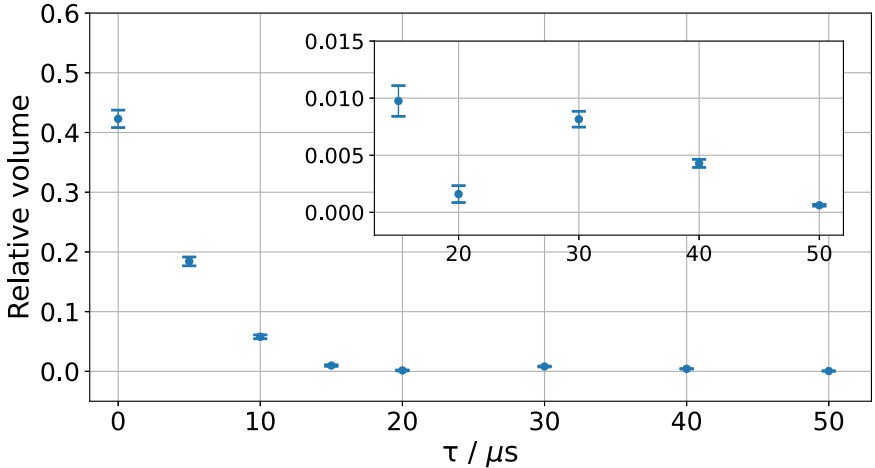

**Fig. 8 | Signature of non-Markovian time evolution at late times.** Numerically determined relative volumes of the realized to consistent state spaces $\mathrm{Vol}(\mathcal{S}^\tau)/\mathrm{Vol}(\mathcal{S}_{\mathrm{consistent}})$ for different waiting times $\tau$, again with the standard deviation determining error bars. The volume always decreases, as required by Markovian time evolution, except between $\tau_4 = 20\,\mu s$ and $\tau_5 = 30\,\mu s$.

typically noncontextual too. Indeed, we have obtained $r = 0$ for $\tau \geq 15\,\mu s$ in all repetitions.

It is well-known that the quantum bit is noncontextual in the sense of Kochen and Specker[2], and that it admits of simple hidden-variable models, including one that has already been given by Bell[41]. However, as shown by Spekkens[3], all such models must be contextual according to the generalized notion introduced in Results' subsection "Theory-independent analysis". As we have demonstrated above, the same is true for the "noisy qubits" that describe the superconducting system for sufficiently small evolution times.

### Non-Markovianity at late times

In Fig. 6, it can be seen that the state space shrinks during all time steps, except between $\tau_4 = 20\,\mu s$ and $\tau_5 = 30\,\mu s$, where it appears to expand. This is a signature of non-Markovianity. Let us define what this means for arbitrary GPTs, including but not restricted to quantum theory. We say that a system $S$ has Markovian time evolution from time $\tau_0$ to $\tau_1 > \tau_0$ if there is a transformation $T$, defined on the system $S$ alone, such that $\boldsymbol{s}_S(\tau_1) = T\boldsymbol{s}_S(\tau_0)$, with $\boldsymbol{s}_S(\tau)$ the state of $S$ at time $\tau$ (in QT; this would be a trace-preserving completely positive map). This requirement is

automatically fulfilled if we are able to prepare the system at time $\tau_0$ in some arbitrary state of $\mathcal{S}$, uncorrelated with other systems and the environment. Mathematically, the corresponding transformation $T$ will then be a linear map on the set of unnormalized states (a well-known fact for GPTs). Thus, it must act as an affine-linear map on the set of normalized states $\mathcal{S}$. Since it maps states to valid states, it holds that $T(\mathcal{S}) \subset \mathcal{S}$, and so it must be volume-non-increasing (see Methods' subsection "Markovian evolutions do not increase the volume" for details).

On the other hand, if there are initial correlations between the system $S$ and its environment $E$ at time $\tau_0$, then initial state $\boldsymbol{s}_{SE}$ and final state $\boldsymbol{s}'_{SE}$ are related by some transformation of the total system $SE$, $\boldsymbol{s}'_{SE} = T_{SE}\boldsymbol{s}_{SE}$. Thus, there will not, in general, be a transformation (or other linear map) $T$ that relates the possible initial and final marginal states $\boldsymbol{s}'_S$ and $\boldsymbol{s}_S$. In the quantum case, this fact has been pointed out by Schmid et al.[42]. Intuitively, this sort of non-Markovianity happens if there is some information backflow from the environment to the system.

We have numerically determined the volumes of the state spaces $\mathcal{S}^\tau$ (see Fig. 8), and see that the volume increases between times

$\tau_4 = 20\,\mu s$ and $\tau_5 = 30\,\mu s$, as is visible in Fig. 6. In particular, for each $\tau$, we calculated the volume of the realized state space $\mathrm{Vol}(\mathcal{S}^\tau)$ relative to the volume of the consistent state space $\mathrm{Vol}(\mathcal{S}_{\mathrm{consistent}})$. The observation that the relative volume increases between $\tau_4$ and $\tau_5$ proves that there is non-Markovian evolution during this time interval. This is physically not unexpected, since the transmon is only approximately a qubit, and higher-lying levels cannot be perfectly ignored. For the experiment we consider, the non-Markovianity may thus be due to the residual off-resonant coupling with an additional degree of freedom, such as another qubit hosted in the same readout cavity resonator. On the other hand, successful state preparation at time $\tau_0 = 0$ implies that time evolution from time $\tau_0 = 0$ to any other time $\tau_1 = \tau$ must always be Markovian, and so $\mathrm{Vol}(\mathcal{S}^0) \geq \mathrm{Vol}(\mathcal{S}^\tau)$ for all $\tau \geq 0$, which is confirmed by our data.

We fit an exponential decay function $Ae^{-\frac{\tau}{B}}$ to the relative volumes for different $\tau$ (weighted by the uncertainties in the relative volume), obtaining values $A = 0.45 \pm 0.01$ and $B = 0.227 \pm 0.004\,\mu s^{-1}$. This provides a theory-independent estimate of the relaxation rate $\frac{1}{B} = 4.409\,\mu s$.

The two phenomena discussed in this and the previous subsection have very different roles to play: generalized contextuality is a precious resource that is hard to obtain; non-Markovianity, on the other hand, is a frequent statistical phenomenon that is often detrimental rather than desirable. What we claim to be remarkable is not that there is non-Markovianity in the experiment, but that we are able to demonstrate it as a property of the system's time evolution without assuming the validity of quantum theory.

## Discussion

Theory-agnostic tomography for preparations and measurements was introduced in ref. 10, where it was applied to a photonic qubit, and subsequently in ref. 11 to a photonic qutrit. The present work makes use of the methods of refs. 10,11 to construct the state and effect space of the best fit GPT model of the superconducting qubit.

Following ref. 11, we determine the best rank fit as the one which least overfits the data, requiring fewer assumptions than the AIC criterion used in ref. 10. However, unlike ref. 11, the 10 different frequency tables obtained provide us with 90 pairs of training and test data, allowing us to give error bars for the best fit $\chi^2$ values for each dimension.

In the present work, we extend theory-agnostic tomography of refs. 10,11 to include a form of process tomography. We use this for theory-independent monitoring of decoherence, and to obtain a theory-independent witness of non-Markovianity. We also monitor the time evolution of generalized contextuality of the superconducting qubit, showing that it is contextual for times $\tau = 0, 5, 10\,\mu s$ and non-contextual for times greater than $15\,\mu s$.

An experimental demonstration of Kochen-Specker contextuality in a three-level superconducting system was shown in ref. 9. The notion of generalized contextuality used in the present work is weaker than Kochen-Specker contextuality in that it does not require projective measurements (but rather applies to POVMs more generally) nor does it require outcome-determinism. Outcome-determinism is the requirement that outcomes of measurements depend deterministically on the underlying hidden variable $\lambda$. As argued in ref. 3, outcome-determinism is not entailed by non-contextuality, and thus should be viewed as an independent assumption in proofs of contextuality.

In addition to the assumption of outcome-determinism/projective measurements, a number of additional assumptions are needed in the demonstration of contextuality of ref. 9, as specified in ref. 9, Supplementary Material. We briefly comment on these assumptions (1.(a), 1.(b), 2. and 3.) and how they relate to the present work. Assumptions 2. and 3. are concerned with sequential measurements and compatibility of measurements, neither of which are needed in

present work. Assumption 1.(a) concerns the probability of picking a particular context, which is not needed in this work either. Assumption 1.(b), which states that the probability distribution over the ontic states is the same in every run, is shared with the present work.

As discussed in section "Contextuality of GPTs", we make the assumption that the preparations and measurements implemented in the experiment are tomographic relative to one another, which is not needed in ref. 9. The tomography loophole for experimentally demonstrating generalized contextuality can be viewed as analogous to the finite precision measurement loophole[43] for experimental demonstrations of Kochen-Specker contextuality.

Finally, we briefly contrast the theory-independent approach based on GPTs used here to the device-independent (DI) approach[44], which can also used to certify nonclassicality using minimal assumptions. In both the DI approach and GPT tomography, the basic objects are frequencies/probabilities over outputs given some inputs, which together with some well-motivated principles form the basis of the analysis. We note that the DI approach is not necessarily theory-independent, for instance, self-testing protocols verify that a certain quantum state (up to a family of local transformations) has been prepared based solely on the Bell inequality violation of the observed statistics[45]. The DI approach typically uses assumptions about causal structure (motivated by special relativity), whereas the GPT tomography approach makes both a causal assumption ($\lambda$-mediation[21]) and a tomographic completeness assumption, which are not motivated by appeal to another physical theory.

To summarize, in this work we have monitored the decoherence of a physical system, its evolution of generalized contextuality, and the (non-)Markovianity of its time evolution in a theory-independent way. This means that we have not assumed the validity of quantum theory in any of the data analysis or the conclusions that we have drawn. All that we had to assume was that the experimental setup includes a physical system that is probed in a way such that the implemented preparations and measurements are tomographically complete for each other, which allowed us to use the GPT formalism that does not depend on any specific choice of theory.

Our analysis has shown us that our experimental data is best described by a GPT that has a four-dimensional space of states and effects (consistent with the description as a qubit). We have shown that the state space shrinks over time (decoherence), loses its contextuality (emergence of classicality), and undergoes non-Markovian evolution at late times. Since we do not need to assume quantum theory in the analysis of the experiment, these properties are demonstrated to hold irrespective of the theoretical description of the physical system. That is, even if quantum theory were to be overturned by another theory in the future, our conclusions would still hold, as long as our experiment and its analysis have been correctly performed, and our assumption of tomographic completeness (see Methods' subsection "Contextuality of GPTs") is satisfied. This is to some extent similar to experimental demonstrations of the violation of Bell inequalities[46–48], which show the failure of local realism not only within quantum theory, but as a property of nature that will survive all future revisions of our theoretical description. While we do not claim the same sort of device independence as for Bell experiments (mainly due to the assumption of tomographic completeness), we believe that our results represent a significant step forward towards the goal of a theory-independent reevaluation of experiments under minimal assumptions, which includes subjecting quantum theory to rigorous scrutiny.

It is instructive to be slightly more specific about what the GPT system really is that we have determined. Our experimental choice of 100 preparation and measurement procedures defines an operational theory, to which we have fitted an effective GPT system with state

space $\mathcal{S}$ and effect space $\mathcal{E}$. As discussed in the Methods' subsection "Determining the set of consistent states", the set of all possible preparations and measurements on the superconducting qubit is larger, with state and effect spaces $\mathcal{S}_{\text{phys}} \supset \mathcal{S}$ and $\mathcal{E}_{\text{phys}} \supset \mathcal{E}$. Due to our assumption of tomographic completeness, however, our effective GPT is, in the terminology of ref. [49], a *fragment* of the physical GPT of the superconducting system. Hence, the proof of generalized contextuality for the former (for times 10 μs or less) applies directly to the latter, and so does the proof of non-Markovianity and the observation that the state space contracts over time.

Our work raises several interesting questions. For example, what can such experiments tell us if we do not make any assumption of tomographic completeness, such as in cases where many-body systems are probed with coarse-grained or collective measurements only[50]? Some progress has recently been made on this question[51], in particular by showing that a notion of *relative tomographic completeness* is sufficient for demonstrating generalized contextuality, but there are still important open questions on how to apply this to the analysis of concrete experiments. Furthermore, can (standardly performed) measurements of the Wigner function, interpretable as large collections of preparation or measurement procedures, reveal generalized contextuality or other phenomena related to the GPT representation[37]? We believe that further theoretical analyses and experimental exploration may lead to interesting new insights into the foundations and practical certification of nonclassicality, as well as novel precision tests of quantum theory.

## Methods

To complement the methods section, in the repository[52] we provide an open-source Python code and the actual data used in this paper.

### Generalized probabilistic theories

A GPT system (without transformations) is described by a convex set of (normalized) states $\mathcal{S} \subset V$ ($V \simeq \mathbb{R}^d$) and a convex set of effects $\mathcal{E} \subset V^*$ such that $0 \leq \langle e, s \rangle \leq 1$ for all states and effects. The set of effects contains the $\mathbf{0}$ effect: $\langle \mathbf{0}, s \rangle = 0$ for all $s \in \mathcal{S}$ and the unit effect $\mathbf{u}$: $\langle \mathbf{u}, s \rangle = 1$ for all $s \in \mathcal{S}$. The number $\langle e, s \rangle$ gives the probability that the measurement outcome represented by $e$ occurs when the system is prepared in state $s$. Given a set of effects $\{e_1, \ldots, e_n\}$ and a set of states $\{s_1, \ldots, s_m\}$, the associated probability table $D$ is a $m \times n$ matrix with entries $D_{ij} = \langle e_j, s_i \rangle$. Two GPT systems $(\mathcal{S}, \mathcal{E})$ and $(\mathcal{S}', \mathcal{E}')$ are *equivalent* if exists an invertible linear map $L : V \to V'$ such that $L(\mathcal{S}) = \mathcal{S}'$ and $(L^{-1})^*(\mathcal{E}) = \mathcal{E}'$, where $T^*$ denotes to adjoint of a linear map $T$. Therefore two equivalent systems yield the same probabilities: $\langle e', s' \rangle = \langle (L^{-1})^*(e), L(s) \rangle = \langle e, L^{-1}L(s) \rangle = \langle e, s \rangle$.

As a simple example of a GPT system, consider a qubit in quantum theory. The normalized states of the qubit are the $2 \times 2$ density operators, which form a convex subset of the real vector space of Hermitian operators on $\mathbb{C}^2$. The effects of a qubit are the positive semidefinite operators $E$ such that $0 \leq E \leq \mathbf{1}$, where 0 is the zero effect and $\mathbf{1}$ the unit effect. The probability of effect $E$ given state $\rho$ is given by $\langle E, \rho \rangle = \text{Tr}(\rho E)$.

The vector representation of a qubit is given by the Bloch representation. The state of a qubit can be written in the form $\rho = \frac{1}{2}\left(\mathbf{1} + \sum_{i=1}^{3} a_i \sigma_i\right)$, where the $\sigma_i$ are the Pauli matrices, which together with the identity $\mathbf{1}$ form a basis for the 4-dimensional real vector space of Hermitian operators on $\mathbb{C}^2$. Positive semidefiniteness of is equivalent to $\sum_i a_i^2 \leq 1$. Thus, the states of a qubit can be equivalently expressed in Bloch vector form as $s_\rho = \frac{1}{2}(1, a_1, a_2, a_3)^\top$ with $\|a\| \leq 1$. The effects can be expressed in the same basis as a vector $e_E = (c_0, c_1, c_2, c_3)^\top$, where $c_i = \frac{1}{2}\text{Tr}(E\sigma_i)$. Hence, the outcome

probabilities are not w $p(0|P_i, M_j) = \text{Tr}(\rho_i E_j) = s_i^\top e_j$. The full probability table can now be written as:

$$D = \begin{pmatrix} s_1^\top \\ s_2^\top \\ \vdots \\ s_m^\top \end{pmatrix} \begin{pmatrix} e_1 & e_2 & \cdots & e_n \end{pmatrix} = SE. \tag{8}$$

Thus, the $m \times n$ probability table $D$ for the qubit can be factored into a $m \times 4$ matrix $S$ and a $4 \times n$ matrix $E$. This implies that $D$ is a rank-4 matrix.

Another canonical example of a GPT system is the $d$-dimensional classical system $\Delta_d$. The normalized states of $\Delta_d$ consist of probability distributions over $d$ outcomes, which form a convex subset of $\mathbb{R}^d$. The effects are the response functions, namely all linear functionals $e$ which map states to $[0, 1]$. In the vector representation, the normalized states are $d$-dimensional vectors with entries in $[0, 1]$ which sum to 1 and the effects are vectors with entries in $[0,1]$. The 0 effect is the vector $(0, \ldots, 0)^\top$ and the unit effect is the vector $(1, \ldots, 1)^\top$. The states form a $d$-dimensional simplex, whilst the effects form a $d$-dimensional hypercube.

### Uniqueness of the decomposition $D = SE$

**Lemma 1**. Suppose that $D = SE$, where $S$ is a real $m \times k$ matrix, $E$ is a real $k \times n$ matrix, and $k = \text{rank } D$. If $D = S'E'$ is another decomposition with these properties, then there is an invertible matrix $L$ such that $S' = SL$ and $E' = L^{-1}E$.

**Proof.** If $M$ is some matrix, we denote its Moore-Penrose pseudoinverse[53] by $M^+$. Set $L := E(E')^+$, then

$$SL = SE(E')^+ = D(E')^+ = S'E'(E')^+. \tag{9}$$

Now, the $k$ rows of $E'$ are linearly independent, because otherwise, we would have $\text{rank } D \leq \text{rank } E' < k$. This implies $E'(E')^+ = \mathbf{1}$, and hence $SL = S'$. Similarly, the $k$ columns of $E$ are linearly independent, which implies both $E^+E = \mathbf{1}$ and $(E(E')^+)^+ = E'E^+$. Thus

$$L^+E = (E(E')^+)^+ E = E'E^+E = E'. \tag{10}$$

It remains to be shown that $L$ is invertible. Using all of above, this follows from

$$L^+L = (E(E')^+)^+ E(E')^+ = E'E^+E(E')^+ = \mathbf{1}, \tag{11}$$

hence $L^+ = L^{-1}$.

### Solving the weighted low-rank approximation problem of Eq. (4)

In this section, we outline the methodology introduced in ref. [10] to find a rank-$k$ matrix $D^{\text{realized}}$, which serves as the best estimate for the ideal probability table $D$ leading to the experimentally observed statistics $F$. Let us recall that this is achieved by minimizing the weighted $\chi_k^2$ statistics in the following optimization problem (4):

$$\min_{D \in M_{mn}} \chi_k^2 = \sum_{ij} \frac{(F_{ij} - D_{ij})^2}{\Delta F_{ij}^2} \tag{12}$$

subject to $\text{rank}(D) = k$, $0 \leq D_{ij} \leq 1$,

where $\Delta F_{ij}^2$ is the statistical variance in $F_{ij}$. Note that $D = SE$ contains the inner products $s_i^\top e_j$, which makes Eq. (4) a non-convex optimization

problem. Moreover, this minimization problem is known to not have an analytical solution[54,55]. Following ref. 10, we employ an iterative approach that decomposes the optimization into two convex subproblems. Specifically, we alternate between optimizing the states while fixing the effects and optimizing the effects while fixing the states, with each optimization being a convex quadratic program[10]. This iterative process, known as the see-saw algorithm[56,57], continues until the cost function, here the $\chi_k^2$, converges to a desired numerical precision. However, it is important to note that due to the non-convexity of the general problem, this procedure does not guarantee convergence to a global minimum. We refer the reader to ref. 10 [Appendix C] for an explicit description of how the $\chi_k^2$ minimization in Eq. (4) can be translated into a series of alternating convex optimizations.

The output of the optimization algorithm is a rank $k$ matrix $D$ with entries in $[0, 1]$. In order to obtain the corresponding GPT states and effects, we must factor $D$ into two matrices $S$ and $E$. Note that this factorization is non-unique. As in ref. 10, we first append a column of 1's to $D$, which reflects the probability associated with the unit effect (corresponding to the identity operator in quantum theory), assuming no experimental losses. Then, as described in ref. 10 [Appendix C], we perform a QR decomposition of $D$ to express it as $D = SE$. This ensures that the first column of $S$ consists of 1's, thus effectively encoding the normalization of the GPT states (analogous to the trace degree of freedom for quantum states).

Hence the matrices of states and effects have the following forms:

$$
S = \begin{pmatrix}
1 & s_1^1 & \dots & s_1^{k-1} \\
1 & s_2^1 & \dots & s_2^{k-1} \\
\vdots & \vdots & \ddots & \vdots \\
1 & s_m^1 \dots & \dots & s_m^{k-1}
\end{pmatrix} = \begin{pmatrix}
\boldsymbol{s}_1^\top \\
\boldsymbol{s}_2^\top \\
\vdots \\
\boldsymbol{s}_m^\top
\end{pmatrix}, \tag{13}
$$

$$
E = \begin{pmatrix}
1 & e_1^0 & \dots & e_n^0 \\
0 & e_1^1 & \dots & e_n^1 \\
\vdots & \vdots & \ddots & \vdots \\
0 & e_1^{k-1} & \dots & e_n^{k-1}
\end{pmatrix} = \begin{pmatrix} \boldsymbol{e}_1 & \boldsymbol{e}_2 & \dots & \boldsymbol{e}_n \end{pmatrix}. \tag{14}
$$

From the optimization problem and choice of factorization, we obtain the unit effect $\boldsymbol{1} = \boldsymbol{e}_0$ and $n$ effects $\boldsymbol{e}_1 \dots \boldsymbol{e}_n$, where $\boldsymbol{s}_i^\top \boldsymbol{e}_j \in [0, 1]$ for all $i, j$ by construction. Every measurement $M_j$ has two outcomes 0 and 1 with $p(0|P_i, M_j) = \boldsymbol{s}_i^\top \boldsymbol{e}_j$. Since $P(0|P_i, M_j) + P(1|P_i, M_j) = 1$ it follows that $p(1|P_i, M_j) = 1 - s_i^\top e_j = s_i^\top (\boldsymbol{1} - e_j)$.

Hence, in order to account for the probabilities $p(1|P_i, M_j)$, the effect space should include the complement effect $(\boldsymbol{1} - \boldsymbol{e}_j)$ for every $\boldsymbol{e}_j$. We append the $(n + 1) \times k$ matrix of complement effects $E'$ with columns $(\boldsymbol{1} - \boldsymbol{e}_j)$ to the matrix $E$ to obtain a $2(n + 1) \times k$ matrix of effects (which we shall label as $E$ hereon for convenience). Augmenting the set of effects in this manner does not change the set of consistent states or effects.

## Determining the best fit rank $k$

When solving Eq. (4), note that as the chosen rank $k$ increases, the error of the fit $\chi_k^2$ naturally decreases. However, note that in general the experimental noise causes $F$ to be a full rank matrix, even when the underlying $D$ it approximates is not. Therefore, increasing $k$ can result in a model that overfits the experimental data, which is to say that it fits the noise in the training set data $F$. In practice, this can be determined by separating the data into a training set $F = F^{\text{train}}$ and a test set $F' = F^{\text{test}}$, where the $\chi_k^{2\,\text{test}}$ is evaluated according to Eq. (5). The $\chi_k^{2\,\text{train}}$ for the training data determines how much the model underfits the data; the higher the $\chi_k^2$ value the worse the fit is. To test for overfitting, the model is applied to the test set $F^{\text{test}}$. Models that overfit will have a

significantly worse $\chi_k^{2\,\text{test}}$ since they will fit to noise that was present in $F^{\text{train}}$ but is not present in $F^{\text{train}}$. There is a value $k^{\text{opt}}$ for which the models will transition from underfitting to overfitting. Then, the matrix $D_{k^{\text{opt}}}$ is chosen as the best fit.

In summary, to estimate the rank $k^{\text{opt}}$, we hypothesize several values $k$ for $k^{\text{opt}}$ and test each one. Then, for each hypothesized rank, we compute the optimal estimate of the rank-$k$ matrix $D$ and evaluate its fit to the data using the $\chi^2$ statistic. Additionally, to decide which $k$ provides the best balance between fitting the experimental data and maintaining simplicity in the model, we split the analysis with a training and a test set to ensure that the model is not overfitting the data. As we have seen in subsection "Generalized probabilistic theories", if the experiment is described by a qubit quantum model (as we expect, since we have prepared and probed a system that is, from quantum physics and the experimental setup, expected to be a qubit), then the rank is expected to be 4. Indeed, following this method, we have shown in Fig. 4a that this is the case. Nonetheless, recall that we are keeping it general to follow a theory-agnostic approach, where we obtain the best GPT fit without assuming the correctness of quantum theory.

## Determining the set of consistent states

The methodology so far estimates one possible set of states and effects compatible with the experimental data, which we denote $\mathcal{S}$ and $\mathcal{E}$. However, as in ref. 10, we can further estimate the set of all logically consistent states and effects for the experiment, named respectively $\mathcal{S}_{\text{consistent}}$ and $\mathcal{E}_{\text{consistent}}$. The former is given by the set of all possible vectors $\boldsymbol{s}$, normalized such that $\langle \boldsymbol{u}, \boldsymbol{s} \rangle = 1$, which, when acted on by any covector of the realized effect space, give valid probabilities, i.e. all $\boldsymbol{s}$ such that $0 \leq \langle \boldsymbol{e}, \boldsymbol{s} \rangle \leq 1$ for all $\boldsymbol{e} \in \mathcal{E}$. Conversely, the latter is defined by the set of all covectors $\boldsymbol{e}$ that take the vectors of the realized state space to valid probabilities, i.e. all $\boldsymbol{e}$ such that $0 \leq \langle \boldsymbol{e}, \boldsymbol{s} \rangle \leq 1$ for all $\boldsymbol{s} \in \mathcal{S}$. These inequalities tell us that the consistent spaces are the *duals* of the realized spaces, i.e., $\mathcal{S}_{\text{consistent}} = \text{dual}(\mathcal{E})$ and $\mathcal{E}_{\text{consistent}} = \text{dual}(\mathcal{S})$. If the realized state space $\mathcal{S}$/effect space $\mathcal{E}$ is smaller (in particular, if it is a strict subset of the set of states $\mathcal{S}_{\text{phys}}$/effects $\mathcal{E}_{\text{phys}}$ that could in principle be implemented on the system, which is always the case in experiments on quantum systems with infinitely many pure states), then its corresponding dual is larger − since there is a larger set of covectors/vectors that combine appropriately to give valid probabilities. Accordingly, we know that $\mathcal{S}_{\text{phys}}$ and $\mathcal{E}_{\text{phys}}$ must lie somewhere between the realized and consistent state/effect spaces, comprising lower and upper bounds, respectively.

The duals are calculated via the `cdd` library in Python[58], and amounts (in their terminology) to converting from the H-representation of a convex polyhedron $P$ to its V-representation. For a given matrix of points, say that of $\mathcal{S}$, that must be combined with $\mathcal{E}_{\text{consistent}}$ to give valid probabilities, a convex polyhedron of the form $P = \{x | Ax \leq b\}$ can be defined, where $A = [-\mathcal{S}], [\mathcal{S}]]^\top$ and $b = [[0]^m, [1]^m]^\top$. The H-representation is the matrix $[b \;\; - A]$, representing the set of linear inequalities characterizing $P$. The polyhedron can also be represented by the convex hull of its vertices, i.e., $P = \text{conv}(v_1, \dots, v_n)$, which is called its V-representation. We can use the `get_genera-tors()` method of the `cdd` library to obtain the V-representation $[t \;\; - V]$, where $t = [[1]^n]^\top$ and $V = [\mathcal{E}_{\text{consistent}}]$.

## Reparametrization of the state space

In order to compare the different $\mathcal{S}^\tau$ for different decoherence times $\tau$, we are interested in finding an appropriate parametrization for these distinct sets. This is achieved by applying a linear transformation that approximates their respective $\mathcal{S}_{\text{consistent}}$ set to a unit sphere. Note that $\mathcal{S}_{\text{consistent}}$ is a function of $\mathcal{E}$, which does by construction not depend of $\tau$. As explained in Subsection "Generalized probabilistic theories", we can always reparametrize GPT systems linearly without changing any

of their relevant properties. The method we present is split in three steps: (1) we make sure that the set of data points we work with are all extremal points; (2) we center the boundary on the origin, and (3) we pose the search for the suitable linear transformation as an optimization problem. After finding an appropriate centering and transformation for the boundary of the consistent state space $\mathcal{S}_{\text{consistent}}$, we apply the same to all realized state spaces $\mathcal{S}^\tau$, so they can be appropriately compared.

Let us start with a cautionary preliminary step by removing all the interior points of a given set of states $\mathcal{S}$. This is to guarantee that we are left only with boundary points. We do so by probing whether a given state $\boldsymbol{s}_i \in \mathcal{S}$ can be obtained as a convex combination of all the other points $\boldsymbol{s}_j \in \mathcal{S}\backslash\{\boldsymbol{s}_i\}$ for $i \neq j$. This can be posed as a feasibility problem using linear programming. If the linear program is feasible, then we know that the state being probed $\boldsymbol{s}_i$ is a mixture of the other points; i.e., there exists a convex combination $\sum_{j \neq i}\lambda_j \boldsymbol{s}_j = \boldsymbol{s}_i$ with $\boldsymbol{s}_j \in \mathcal{S}\backslash\{\boldsymbol{s}_i\}$, $0 \le \lambda_j \le 1$ and $\sum_{j \neq i}\lambda_j = 1$. The feasibility linear program is the following:

$$\text{find } \boldsymbol{\lambda}$$
$$\text{subject to } \sum_{j \neq i}\lambda_j \boldsymbol{s}_j = \boldsymbol{s}_i, \quad \sum_{j \neq i}\lambda_j = 1, \quad 0 \le \lambda_j \le 1 \text{ for all } j \neq i. \quad (15)$$

Therefore, when the linear program is feasible, we discard the state $\boldsymbol{s}_i$ and update the set of states to $\mathcal{S}' = \mathcal{S}\backslash\{\boldsymbol{s}_i\}$. We iterate this procedure for all $i \in \{1, \ldots, n\}$ until we have tested all states $\boldsymbol{s}_i$ and we are left only with the ones which cannot be obtained as convex combinations of the rest.

Next, we center the boundary $\mathcal{S}'$ on the origin by subtracting the columnwise mean; namely $\mathcal{S}_{\text{centered}} = \mathcal{S}' - \boldsymbol{\mu}$, where is a $\boldsymbol{\mu}$ is a column vector with entries $\boldsymbol{\mu}_j = \frac{1}{n}\sum_{i=1}^{n}\mathcal{S}_{i,j}$ denoting the mean for column $j$ of $\mathcal{S}'$.

Finally, let us now provide a method to find a linear transformation that maps $\mathcal{S}_{\text{centered}}$ closer to the unit sphere. Consider a state $\boldsymbol{s}_i \in \mathcal{S}_{\text{centered}}$; the idea is to find a transformation $L$ such that $\| \boldsymbol{s}'_i \| \approx 1 \forall i$ with $\boldsymbol{s}'_i := L\boldsymbol{s}_i$. If we focus in the case where the state space has rank $k = 4$, then $L$ will be a $(k-1) \times (k-1) = 3 \times 3$ matrix. By means of the singular value decomposition, $L = U\Sigma V^\top$, note that we are only interested in finding a diagonal matrix $\Sigma := \text{diag}(\sigma_1, \sigma_2, \sigma_3)$ with $\sigma_i \in \mathbb{R}_{\geq 0}$ and a $3 \times 3$ real orthogonal matrix $V$. We can neglect $U$ since, by definition, $\| \boldsymbol{s}'_i \| = \sqrt{\boldsymbol{s}_i^\top V \Sigma^\top U^\top U \Sigma V^\top \boldsymbol{s}_i} = \sqrt{\boldsymbol{s}_i^\top V \Sigma^2 V^\top \boldsymbol{s}_i}$. Furthermore, we use Euler angles parametrization for $V$, which guarantees the constraint $VV^\top = \mathbf{1}$ while having full coverage of SO(3) with 3 parameters (angles) $\alpha, \beta, \gamma \in \mathbb{R}$:

$$V := \begin{pmatrix} \cos(\alpha)\cos(\gamma) - \sin(\alpha)\cos(\beta)\sin(\gamma) & -\cos(\alpha)\sin(\gamma) - \sin(\alpha)\cos(\beta)\cos(\gamma) & \sin(\alpha)\sin(\beta) \\ \sin(\alpha)\cos(\gamma) + \cos(\alpha)\cos(\beta)\sin(\gamma) & -\sin(\alpha)\sin(\gamma) + \cos(\alpha)\cos(\beta)\cos(\gamma) & -\cos(\alpha)\sin(\beta) \\ \sin(\beta)\sin(\gamma) & \sin(\beta)\cos(\gamma) & \cos(\beta) \end{pmatrix}. \quad (16)$$

Therefore, to find the suitable $\Sigma$ and $V$, we consider the following constrained minimization problem:

$$\min_{\sigma_1, \sigma_2, \sigma_3, \alpha, \beta, \gamma \in \mathbb{R}} \sum_{i=1}^{n}\left(1 - \| \boldsymbol{s}'_i \|^2\right)^2$$
$$\text{subject to } \Sigma := \text{diag}(\sigma_1, \sigma_2, \sigma_3), \quad (17)$$
$$\sigma_1, \sigma_2, \sigma_3 \ge 0.$$

Once we find the suitable translation $\mu$ and transformation $L = \Sigma V^\top$ for a given $\mathcal{S}_{\text{consistent}}$, we apply them in turn to all realized state spaces $L(\mathcal{S}^\tau - \mu)$ so that we can properly compare the sets as shown in Fig. 6.

## Lower bound on the purity of the prepared states

If we assume that quantum theory holds, then there will be a valid qubit density matrix $\rho_s$ for every state $\boldsymbol{s} \in \mathcal{S}$. Let us assume that $\lambda_{\max}(\rho_s) \le c$ for all $\boldsymbol{s} \in \mathcal{S}$, where $\frac{1}{2} \le c \le 1$. Then the Bloch vectors of all $\rho_s$ are contained in ball of radius $2c - 1$ around the origin in the Bloch representation. Since the one-norm distance of quantum states corresponds to the Euclidean distance in the Bloch ball, this means that $\mathcal{D}(\boldsymbol{s}, \boldsymbol{s}') \le \frac{1}{2}\|\rho_s - \rho_{s'}\|_1 \le 2c - 1$ for all $\boldsymbol{s}, \boldsymbol{s}' \in \mathcal{S}$. Hence $c \ge \frac{1}{2}\left(1 + \max_{\boldsymbol{s}, \boldsymbol{s}' \in \mathcal{S}}\mathcal{D}(\boldsymbol{s}, \boldsymbol{s}')\right)$, and the distinguishability maximum is numerically determined to be at least $0.691 \pm 0.004$, thus $c \ge 0.846 \pm 0.002$.

## Markovian evolutions do not increase the volume

If $T$ is any transformation on a GPT system, then it acts linearly on the set of unnormalized states in $\mathbb{R}^k$[14,15]. In particular, in the parametrization where the first component of a state vector is its normalization, we have $\boldsymbol{s}' = \begin{pmatrix} 1 \\ s' \end{pmatrix} = T\begin{pmatrix} 1 \\ s \end{pmatrix}$, and so $s' = T's + t$, with some vector $t$ and linear transformation $T'$ on $\mathbb{R}^{k-1}$. (To see this explicitly, write $T = (T_{ij})_{i,j=0}^{k-1}$, then $s'_j = T_{j0} + \sum_{n=1}^{k-1}T_{jn}s_n =: t + T's$.) Clearly, transformations map valid states to valid states, so the image of the full state space must satisfy $T'\mathcal{S} + t \subseteq \mathcal{S}$, and so

$$\text{Vol}(\mathcal{S}) \ge \text{Vol}(T'\mathcal{S} + t) = \text{Vol}(T'\mathcal{S}) = |\det T'|\text{Vol}(\mathcal{S}). \quad (18)$$

Thus, $|\det T'| \le 1$, and hence the action of the transformation on the normalized state space cannot increase the volume of any region.

To compute the volumes shown in Fig. 8 for each $\mathcal{S}^\tau$, we have used the *Delaunay Triangulation*[59] on their boundary points. This splits the polytope into non-overlapping simplices (tetrahedras in our case), for which the volume formula can be easily calculated. The total volume of $\mathcal{S}^\tau$ is then estimated by summing the volumes of these simplices.

## Decoherence in GPTs

In the main text, we say that the state of the superconducting system decoheres, even though the notion of "decoherence" is usually only used in QT. Here, we point out that (1) there is a rigorous definition of decoherence for GPTs in the literature which applies to the long-time behavior of the superconducting system; and (2) for finite times, the system evolves irreversibly, and this allows us to draw very similar conclusions to the quantum version of decoherence, justifying the name also beyond QT.

Regarding (1), we will use the definition of ref. 60, which is slightly more general than that of ref. 61 or ref. 62: a complete decoherence map $T$ is a transformation such that $T^2 = T$, i.e., applying the transformation twice is the same as applying it once. In QT, an example is given by the map that removes all off-diagonal elements of a density matrix. Figure 6 indicates that the long-time effect of the time evolution is to map every initial state $\omega$ to the same final state $\omega_0$ (which we physically expect to be the ground state), $T(\omega) = \omega_0$. This is clearly a decoherence map according to the previous definition, since $T(T(\omega)) = T(\omega_0) = \omega_0$.

To see (2), note that reversible transformations $T$ map pure states of a GPT system to pure states, and they preserve the state space $\mathcal{S}$, i.e., $T(\mathcal{S}) = \mathcal{S}$. This is clearly not the case in our system's time evolution: most pure states are mapped to mixed states (in particular, for times

$\tau \lesssim 15\mu s$), and the image of $\mathcal{S}$ is strictly smaller than $\mathcal{S}$. Hence, we have irreversible time evolution on the superconducting system $S$ that we are probing. However, if we assume that the total evolution of $S$ and its environment $E$ is reversible (in QT, this would be equivalent to global unitarity), then it cannot act on $S$ and on $E$ independently (otherwise, $S$ and $E$ would individually preserve their purity). In other words, it must be the reversible interaction between $S$ and $E$ that makes the initially close to pure state of $S$ mixed. This resembles the phenomenology of decoherence of QT and justifies the use of this term in the more general GPT context.

## Contextuality of GPTs

Having shown the image of the GPT state space according to the time delay $\tau$, we want a theory-independent test of their (non-)contextuality. It was shown in ref. 29 that non-contextuality can be established via a linear program which takes as an input a set of states and a set of effects, and determines whether their statistics, via a specified probability rule, can be reproduced by a classical (noncontextual) model. We make use of an open-source version of this program[30], which, moreover, in the case that the GPT is contextual, computes the amount of noise that must be added such that a noncontextual model can be fitted. More specifically, the linear program of ref. 30 is a test of *simplex embeddability*: the property that a GPT's state space can be embedded into a classical probability simplex, with its effect space as the dual of that simplex. This was shown in refs. 28,49 to be equivalent to the existence of a non-contextual ontological model.

For the nonembeddibility of the GPT to meaningfully imply non-classicality, we must assume that there is some system that our experiment probes for which our preparations and measurements are *relatively tomographically complete*[51]. We note that the system being probed will in general be a subsystem of some larger system, for instance in this case it corresponds to the first two energy levels of an anharmonic oscillator. However, crucially, it must be such that the preparations and measurement of that subsystem are tomographically complete relative to each other, or equivalently, that the subsystem being probed is embeddable within the larger system[60]. Without assuming so, it is always possible that the statistics derive from some higher-dimensional, noncontextual system, c.f. the Holevo projection[63]. Tomographic completeness in itself is a theory-independent assumption, but can be strongly motivated here by quantum physics−in particular, by the state space of the qubit being a Bloch ball, for which we have chosen a tomographically highly over-complete distribution of points to sample ($k = 4$ linearly independent states and effects would have been sufficient for tomographic completeness, but many more of these procedures are needed to obtain an accurate estimate of the shape of the state and effect spaces as depicted in Figs. 5 and 6).

To use the linear program, one inputs the GPT's set of states $\mathcal{S}^\tau$ and set of effects $\mathcal{E}$, which were found via the theory-independent analysis of Results' subsection "Theory-independent analysis", as well as two row vectors specifying the unit effect $\boldsymbol{u}$ and the average state $\boldsymbol{m}^\tau$. The unit effect is given by $(1, 0, 0, 0)$, which is always the first entry of the set of effects, whilst the average state is calculated by taking the columnwise mean of the set of states. The program then outputs a coefficient $r$, termed the *robustness of nonclassicality*, as well as a set of epistemic states and response functions (which specify a noncontextual ontological model for the scenario). The important information is encoded in $r$, which quantifies how much depolarizing noise must be added such that a noncontextual onto-logical model can be fitted. In particular, the maximally mixed state defines a depolarization map $D_r^\tau(\boldsymbol{s}) := (1 - r)\boldsymbol{s} + r\boldsymbol{m}^\tau$; the minimum $r$ such that a noncontextual ontological model can be fitted to the scenario thus constitutes an operational measure of nonclassicality. More specifically, a robustness of $r = 0$ indicates that a noncontextual

ontological model can be constructed precisely, without adding noise, whilst $r > 0$ indicates that some amount of depolarizing noise is necessary for an ontological model−thereby witnessing contextuality.

## Experimental setup and qubit readout characterization

The device used for this work is a superconducting transmon qubit hosted inside a three-dimensional readout cavity resonator. For operation, this device is cooled down at ~ 10 mK temperature by a dilution refrigerator and operated by commercial microwave electronics, see Fig. 9 for an illustration. Qubit rotation and readout pulses are generated at room temperature by an arbitrary waveform generator and then up-converted to gigahertz frequencies. After a room-temperature amplification stage, these pulses are sent to the sample inside the refrigerator through a line of cryogenic attenuators and filters protecting the qubit from undesired electromagnetic radiation. The qubit readout pulse is transmitted through the cavity, amplified by both cryogenic and room-temperature components, down-converted to megahertz frequencies, and finally digitalized by a FPGA before being recorded on a PC.

Following the standard practice in superconducting qubit experiments, the state of our qubit is measured in the strong dis-persive regime of circuit-QED[19]. To calibrate our readout and char-acterize its fidelity, we collect 2000 measurement shots for the qubit

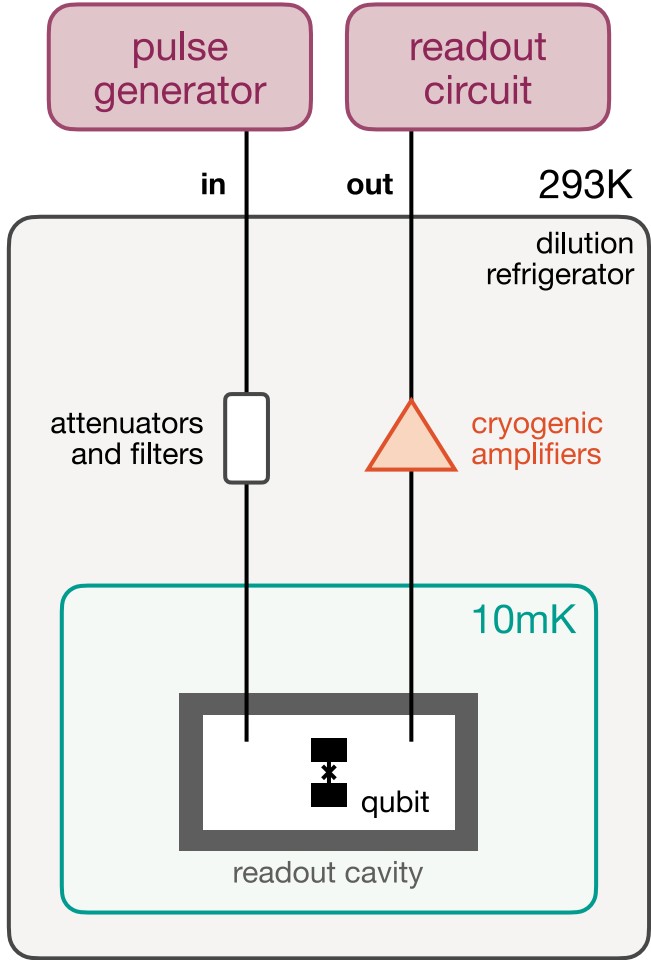

**Fig. 9 | Schematic illustration of the experimental setup used in this work.** The superconducting qubit device used in this work is hosted inside a three-dimensional microwave cavity cooled down at cryogenic temperature by a dilution refrigerator. Qubit state preparation and readout are controlled by microwave electronics at room temperature.

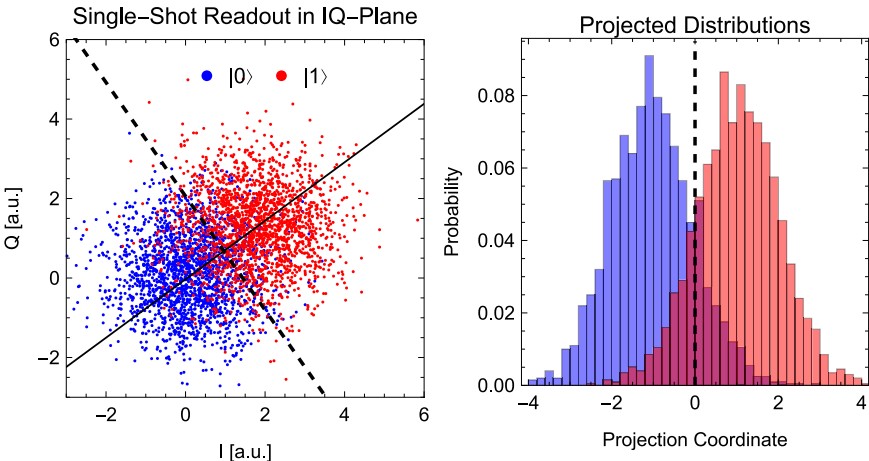

**Fig. 10 | Illustration of the qubit readout characterization.** Left: for each qubit initial state, 2000 measurements are collected for identifying the decision boundary (dashed line) in the IQ-plane. Right: projection of the distributions along the direction connecting their means (solid line).

### Table 1 | Assignment probabilities for prepared quantum states

|  | Assigned \|0⟩ | Assigned \|1⟩ |
|---|---|---|
| Prepared \|0⟩ | 0.846 | 0.154 |
| prepared \|1⟩ | 0.148 | 0.852 |

in its ground state |0⟩ and an equal number of shots for the qubit prepared in |1⟩. Representative results are shown in Fig. 10 (left), which allow us to identify the decision boundary for the |0⟩ vs. |1⟩ state. For clarity, we also show a 1D projection of the measurement distributions along the direction connecting their means in Fig. 10 (right).

From the collected statistics, we can identify the conditional probability $p(a|P)$ of assigning the label $a \in \{|0⟩, |1⟩\}$ when state $P \in \{|0⟩, |1⟩\}$ was prepared. These are given in Table 1.

The average readout fidelity is then defined as $1 - (p(0|1) + p(1|0))/2$, which gives 85(1)%. Contributions to the infidelity come from the finite cavity dispersive shift, qubit state decay during the ~ 10 μs long readout pulse, as well as finite thermal population of ~ $10^{-3}$ at equilibrium.

## Data availability
The data that support the findings of this study are publicly available in the Zenodo repository[52].

## Code availability
The custom code used to produce the results presented in this paper is publicly available in the Zenodo repository[52].

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

## Acknowledgements

T.G. acknowledges helpful discussions with Robert Spekkens and Patrick Daley, and thanks Michael Mazurek for providing him with the data from ref. 10. These contributions helped with writing an initial version of part of the code used in the present paper. This research was funded by the Austrian Science Fund (FWF) 10.55776/PAT2839723; funded by the European Union—NextGenerationEU (AA, TDG, CLJ, MPM). Furthermore, this research was supported in part by Perimeter Institute for Theoretical Physics (MPM). Research at Perimeter Institute is supported by the Government of Canada through the Department of Innovation, Science, and Economic Development, and by the Province of Ontario through the Ministry of Colleges and Universities. M.F. was supported by the Swiss National Science Foundation Ambizione Grant No. 208886, and by The Branco Weiss Fellowship—Society in Science, administered by the ETH Zürich.

## Author contributions

M.F. performed the experiment, A.A., C.J., T.G., and M.M. worked out the theoretical results and wrote the paper, A.A., C.J., and T.G. performed the numerical analysis, and M.M. directed the research.

## Competing interests

The authors declare no competing interests.
