## [Transparent Peer Review file · Nature Communications]

Theory-independent monitoring of the decoherence of a superconducting qubit with generalized contextuality

Corresponding Author: Dr Markus Müller

Version 0:

Reviewer comments:

Reviewer #1

(Remarks to the Author)

I recommend this paper for publication in Nature Communications. The paper applies state-of-the-art theoretical methods to a state-of-the-art experiment, and as such is sure to be a landmark paper in precision experiments testing the nature of quantum reality. I expect works like this one will stimulate a new generation of foundational tests of nonclassicality that come with fewer assumptions and stronger conclusions than prior experiments (such as Ref. 8, which is in some sense a precursor to the present work).

My only concern about the manuscript is the handling of the non-Markovian evolution that was observed. First of all, it seems inappropriate to me that the observation of contextuality and non-Markovianity are rolled together into one section. If I'm understanding things correctly, the observation of contextuality in the present system is one of the strengths of the experiment, while the observation of non-Markovianity is essentially a blemish on the otherwise very elegant and well-characterized experiment and analysis. Do the authors agree with this assessment—that this non-Markovianity was unexpected and not well understood? A speculation is made about where this non-Markovianity is coming from, but it would have been nice if there was some evidence or argument to back up this speculation.

The claim that “As discussed in the Methods section, for all GPT systems, physical transformations cannot increase the volume...” is unclear. Non-Markovianity evolutions are physical, after all. Then, in the methods section where the proof of this claim is given, I could not follow the logic. Maybe I am just being thick, but how does one arrive at $s' = T's+t$? Why does $\text{Vol}(T'S) = |\det T'| \text{Vol}(S)$? Why does $|\det T'| < 1$ follow from what came before, as suggested by the word “Thus”? I would love for this argument to be made more clearly, as it is a simple but interesting conclusion, and one I hadn't seen in the literature before. (Is there a theorem proving the quantum version of this result? If so it would be insightful to point the readers to it.)

Comments and concerns, in order of appearance and not importance:

1. The first two sentences of the Introduction are not really sensible, and presumably don't say quite what the authors intended.
2. I feel the word “tractable” in the introduction would better be “relevant”. The present experiment does not seem more “tractable” than a standard Bell test, but rather more sophisticated, by my estimation.
3. The claim that generalized contextuality (henceforth gC) subsumes Wigner negativity should probably also cite the following paper, as the connection was only rigorously established therein: <https://quantum-journal.org/papers/q-2024-03-14-1283/>
4. The statement “a classical model (also known as an ontological model or hidden variable model)” is confusing, especially in the context of the present paper where the notion of classical-explainability is explicitly not “existence of an ontological model for an operational theory”.
5. The nonstandard term “deterministic classical states” should be changed or clarified/defined.
6. The mixture $\lambda P_1 + (1-\lambda) P_2$ is not strictly meaningful, as procedures do not live in a vector space. (If the authors cannot find any simple way to fix this problem, perhaps it is an acceptable abuse to leave it as is, since it doesn't really matter for the rest of the paper. Or, a footnote could be added to the effect that, rather than mixing procedures, what one must do to formalize this fully is to mix states of knowledge about what procedures were done, as done in, e.g., <https://arxiv.org/abs/2009.03297>.)
7. The citations given for the claim that gC is a resource for information processing seem to me a bit random and incomplete.

At a minimum I think the work(s) linking generalized contextuality to quantum computation should be cited. Many other papers would be appropriate to cite here as well—arguably moreso than some chosen by the authors—but I'll leave it to the authors to handle as they please.

8. The claim that “outcome determinism is not entailed by noncontextuality, and thus should be viewed as an independent assumption in proofs of contextuality” could be strengthened. The relevant point seems to me to be that the assumption isn't really motivated by anything. If giving up determinism let us recover a simple classical (but stochastic) explanation of the world, quantum foundations as a field would not exist. Physicists simply would accept indeterminism. (There was a debate to this effect in the literature on Bell scenarios, where some folks took the conclusion to simply be that one must reject determinism. But I think those folks lost that debate, and I believe that this is widely accepted by now.)

9. The authors do not claim the same sort of device-independence as Bell, but do consider their work a significant step forward towards theory-independent experimental demonstrations. I agree with this, but I think a bit more could be said, or a citation could be given to somewhere that this is discussed more fully, since I think a non-expert reader might not really know how to square these two claims.

10. I do not agree with the claim that “It is instructive to be slightly more specific about what the physical system really is... following 43, 44, one can say that we have probed the properties of the effective physical system that is determined by our experimental setup and choice of 100 preparation and measurement procedures”. Strong criticisms have been made against the perspective that a collection of experimental procedures can meaningfully and generally constitute a notion of a system; see for example the discussion in <https://journals.aps.org/prl/abstract/10.1103/PhysRevLett.132.050202>, where it is shown that if one tries to take this point of view on a system, one generally reaches silly conclusions. I am concerned that this (in my opinion problematic) view keeps getting repeated in the literature, without any defence against the criticism made against it. If the authors are nonetheless sympathetic to this perspective, I encourage them to defend the position explicitly.

11. Progress was made in answering the question of “what can such experiments tell us if we do not make any assumption of tomographic completeness...” in the authors' Ref. 57.

A few more comments are in order. I did not check the experimental details of the work, nor all of the data analysis. However, I did go through the basic methods of the paper, and they are not only sound, but state of the art. Also, I do not believe that the paper introduces any significant new theoretical ideas. However, as I stated above, I think that the experiment will be an inspiration for future experiments that aim to study quantum systems under minimal theoretical assumptions. In addition, the experimental capabilities required for the present experiment seem to me impressive, and perhaps uniquely suited to platforms of this kind. As such, I recommend publication.

(Remarks on code availability)

Reviewer #2

(Remarks to the Author)

Aloy et al analyze data taken on a superconducting circuit experiment from the point of view of generalized contextuality. They claim to have evidence of non-Markovian behaviour and of decoherence without assuming quantum theory.

I find original the idea of measuring a qubit in many different axis to find that a single qubit can exhibit non contextual behaviour. I do find that a discussion about Bell's 1965 paper on the existence of hidden variable theories for single qubits (where he exposes a 30 year old hidden assumption in a proof in von Neuman's book of classical impossibility to explain quantum mechanics) is missing. I also find insightful the idea of measuring the rank and the fluctuation of the fit to the matrix to measure the level of non-contextuality of the probabilities measured.

While the research direction is interesting, the experimental data is of too low quality and badly presented. This is demonstrated by the fact that figure 2 is simulation and not the data, which is readily available to the authors. I can only assume the data did not look good, and confirm this guess by looking at the data presented. This is uncomfortable since the single qubit experiment here performed could have been done with very high quality using sunlight, a set of polarizing sunglasses and a camera addressing the polarization of light as qubit. Decoherence and non-Markovianity could have been simulated and detected by averaging over different preparations.

I also disagree with the claim that "Obtaining these conclusions in a theory-independent way establishes that these are properties of the physical system, and not of our theoretical description of the system." This types of claims must be reserved for more elegant experiments that technically exhaust alternative explanation by control experiments. There is not nearly enough evidence here to discard trivial artifacts.

I cannot recommend this paper for publication.

Minor comments:

- Plots are low quality, pixelated and with overlapping legends.

- I do not feel it is proper to talk about decoherence without assuming quantum theory. Decoherence is only meaningful within quantum theory. Across the paper, the authors do a bad job at drawing the line between quantum ideas and more (or less) general notions regarding probability theory.

- The theory agnostic tomography is very inefficient. Quantum tomography of a single qubit takes only three mean values. So while maybe philosophically important it is not practical. So a good reason to do it (or to pay attention to it) needs to be given

and explained. There is probably some foundational line of thought that justifies the extra effort and will shine new light on quantum theory, but I cannot think of it and the authors do not provide any.

- I am not convinced about the observation of non-Markovianity. Non-Markovianity is not a big deal, many effects can cause that in this experiment. Like resonant coupling with a high-Q mode is a typical happening in these types of experiments. (Why the authors say off-resonant?) In the field of superconducting circuits, spurious two level systems (TLSs) are typically coherent and give non-Markovian effects. They are so common that they impose an important limitation of quantum computations and amount to a large part of the error budget in many-qubit experiments. Now, the authors do not show convincing evidence that they have observed a (mundane) non-Markovian effect. Many Markovian effects can cause an increase of coherence. Imagine a Rabi oscillation under photon loss. When the qubit is close to ground, purity goes up. This may not be what the authors see, but I am just giving it as a trivial example of how increase in purity can be explained by Markovian effects. The point is: the authors have not done nearly enough control experiment to make their grand claims. In this particular case, the author could do control experiments, find that mode or TLS and confirm that what they discovered via a theory agnostic experiment is in fact there. That would be cute, but the claim by itself is badly supported by the analysis and the data.

-"dispersive-readout" was not "mentioned above". They called it "measure by dispersive coupling". No need to be inconsistent with terminology.

- I may be wrong here, but I don't think so: equation (1) is poorly introduced. It assumes a factorization that is not justified by probability theory. I understand it is equivalent to $P(A,B|C) = P(A|C)P(B|C)$. I think I know why the authors write this, it is sadly usually found in this context and known as the Reichenbach factorization principle of common cause (and used by Bell in his original proof, improved by CHSH, in page 116 of Nielsen and Chuang there is a derivation without this factorization, which I deem to be the fundamentally correct one). However, the fundamentally correct factorization is (Bayes) $P(A,B|C) = P(A|B,C)P(B|C)$. They are only equivalent, I understand, if the common cause C (lambda in their notations) makes any of these probabilities be $P=P^2=0,1$. i.e. if C makes the system completely deterministic.

(Remarks on code availability)

Reviewer #3

(Remarks to the Author)

(Remarks on code availability)

Version 1:

Reviewer comments:

Reviewer #1

(Remarks to the Author)

I am happy with the authors' response to my concerns, and recommend their paper for direct publication.

I still did not fully understand the authors' treatment of point 10 in my first review. However, this is a tangential point and one that is slightly contentious in the literature, so I'm happy to leave it as is.

I appreciated the other updates and clarifications.

There is a typo on page 14: "und justifies" should be "and justifies"

I have also read the other referee's concerns, and consider them to have been properly addressed by the authors.

(Remarks on code availability)

Reviewer #2

(Remarks to the Author)

(Remarks on code availability)

Reviewer #3

(Remarks to the Author)

(Remarks on code availability)

I. REVIEW REPORT 1

We would like to thank Reviewer 1 for their helpful comments which have, as we believe, significantly improved our paper. Below, we respond to each of the comments individually. (Reviewer comments in green, answers in black.)

I recommend this paper for publication in Nature Communications. The paper applies state-of-the-art theoretical methods to a state-of-the-art experiment, and as such is sure to be a landmark paper in precision experiments testing the nature of quantum reality. I expect works like this one will stimulate a new generation of foundational tests of nonclassicality that come with fewer assumptions and stronger conclusions than prior experiments (such as Ref. 8, which is in some sense a precursor to the present work).

We are grateful for the positive assessment of our work, and we share the reviewer’s hope for a new generation of foundational tests of nonclassicality.

My only concern about the manuscript is the handling of the non-Markovian evolution that was observed. First of all, it seems inappropriate to me that the observation of contextuality and non-Markovianity are rolled together into one section. If I’m understanding things correctly, the observation of contextuality in the present system is one of the strengths of the experiment, while the observation of non-Markovianity is essentially a blemish on the otherwise very elegant and well-characterized experiment and analysis. Do the authors agree with this assessment—that this non-Markovianity was unexpected and not well understood? A speculation is made about where this non-Markovianity is coming from, but it would have been nice if there was some evidence or argument to back up this speculation.

We agree that it was a bad idea to discuss contextuality and non-Markovianity in a single section. We have thus split it into two separate sections. We also agree that these two phenomena have very different roles to play in general: contextuality is a precious resource that is hard to obtain; non-Markovianity, on the other hand, is an (even classically) frequent statistical phenomenon which is often detrimental rather than desirable.

However, the observation of non-Markovianity in our experiment should not be regarded “a blemish”: the task we set ourselves is to accurately monitor the naturally given time evolution of the superconducting qubit (including the evolution of its contextuality properties), and to do so in a theory-independent way. Our results prove that this system evolves in a non-Markovian way at late times, and we show this *without assuming quantum theory*. This phenomenon is physically to be expected: despite all their advantages, transmon qubits devices are inevitably coupled to their solid-state and electromagnetic environment. In particular, it is rather common to observe spurious couplings to long-lived two-level-system (TLS) defects or off-resonant microwave modes of the cavity in which they are hosted. Our theory-agnostic analysis confirms the presence of non-Markovianity.

Put differently, the experimental challenge here was *not* to maintain coherence for as long as possible, or to ensure an otherwise desirable time evolution of the system. Instead, our goal was to *theory-independently monitor* the natural time evolution and its properties (which includes decoherence, and also non-Markovianity, as it turns out); this led to the challenge to prepare a large number of states reliably, and a large number of times. It also shows the robustness of generalized contextuality, which we could certify despite the fact that the initial states have been rather mixed.

We have added a paragraph at the end of the new non-Markovianity subsection to clarify this point, and we have also added a sentence that mentions additional evidence for our speculation as to where this non-Markovianity is coming from (note however our goal was not to explain the *physical origin* of this non-Markovianity, but its *theory-independent verification*).

The claim that “As discussed in the Methods section, for all GPT systems, physical transformations cannot increase the volume. . .” is unclear. Non-Markovianity evolutions are physical, after all. Then, in the methods section where the proof of this claim is given, I could not follow the logic. Maybe I am just being thick, but how does one arrive at $s' = T's + t$? Why does $\text{Vol}(T'S) = |\det T'| \text{Vol}(S)$? Why does $|\det T'| < 1$ follow from what came before, as suggested by the word “Thus”? I would love for this argument to be made more clearly, as it is a simple but interesting conclusion, and one I hadn’t seen in the literature before. (Is there a theorem proving the quantum version of this result? If so it would be insightful to point the readers to it.)

We agree that the argument presented in the previous version of our manuscript was not fully clear, and we have made an effort to improve its clarity. On the one hand, we have added two paragraphs in (what is now) Subsection E (Non-Markovianity at Late Times), explaining this in more detail, and we have added a sentence to the Methods Subsection H (“To see this explicitly, . . .”). Note that some linear algebra facts are used, such as: if L is a linear map, and S a measurable set, then $\text{Vol}(LS) = |\det L| \text{Vol}(S)$. Unfortunately, we have not found a corresponding quantum version of the $|\det T'| \leq 1$ statement, even though it is clearly visible in the typical pictures of qubit quantum channels depicted for example in the book by Nielsen and Chuang.

Comments and concerns, in order of appearance and not importance:

1. The first two sentences of the Introduction are not really sensible, and presumably don’t say quite what the authors intended.

We agree, and we have changed the first two sentences into the following: *Demonstrating that some quantum systems have properties which genuinely defy classical explanation is at the forefront of current theoretical and experimental research.*

2. I feel the word “tractable” in the introduction would better be “relevant”. The present experiment does not seem more “tractable” than a standard Bell test, but rather more sophisticated, by my estimation.

We agree that our specific experiment is not “tractable” in the sense of “trivial”, but comes with some effort. Nonetheless, we still think it is correct to say that it is “often more tractable” to focus on single rather than composite systems, since this avoids some of the difficulties arising from synchronizing and controlling spacelike separated quantum systems. We have hence added the half-sentence “*avoiding the need for spacelike separation*” to clarify that we refer to tractability in *this* sense, not to general simplicity of the experiment.

3. The claim that generalized contextuality (henceforth gC) subsumes Wigner negativity should probably also cite the following paper, as the connection was only rigorously established therein: <https://quantum-journal.org/papers/q-2024-03-14-1283/>

We have added the citation, which is now reference [9].

4. The statement “a classical model (also known as an ontological model or hidden variable model)” is confusing, especially in the context of the present paper where the notion of classical-explainability is explicitly not “existence of an ontological model for an operational theory”.

Yes, we agree. In four places, we have changed “classical model” to “hidden-variable model”, and it still says that this is also called an “ontological model”.

5. The nonstandard term “deterministic classical states” should be changed or clarified/defined.

Indeed, this is not the best sort of terminology. We have changed “[...] is given by a measurable space Λ corresponding to the set of deterministic classical states” to, simply, “[...] is defined on a measurable space Λ .”

6. The mixture $\lambda P_1 + (1 - \lambda)P_2$ is not strictly meaningful, as procedures do not live in a vector space. (If the authors cannot find any simple way to fix this problem, perhaps it is an acceptable abuse to leave it as is, since it doesn’t really matter for the rest of the paper. Or, a footnote could be added to the effect that, rather than mixing procedures, what one must do to formalize this fully is to mix states of knowledge about what procedures were done, as done in, e.g., <https://arxiv.org/abs/2009.03297>.)

We agree, and we have changed this. We have removed the undefined convex combination of procedures, and instead write:

For our scenario, we may not only consider the actually implemented procedures M_i and P_j , but also statistical mixtures of those, such as the preparation procedure P_{mix} which results in implementing either preparation P_1 or P_2 with probability λ or $1 - \lambda$, respectively. [...] Statistical mixtures are then represented by convex combinations, such that, for example, $\mathbf{s}_{P_{\text{mix}}} = \lambda \mathbf{s}_{P_1} + (1 - \lambda) \mathbf{s}_{P_2}$.

7. The citations given for the claim that gC is a resource for information processing seem to me a bit random and incomplete. At a minimum I think the work(s) linking generalized contextuality to quantum computation should be cited. Many other papers would be appropriate to cite here as well—arguably moreso than some chosen by the authors—but I’ll leave it to the authors to handle as they please.

On second thought, we agree with this assessment. We have removed one of our citations (Duarte-Amaral), and have added two further ones, including one that links gC to quantum computation:

D. Schmid and R. W. Spekkens, *Contextual Advantage for State Discrimination*, Phys. Rev. X **8**, 011015 (2018).

D. Schmid, H. Du, J. H. Selby, and M. F. Pusey, *Uniqueness of Noncontextual Models for Stabilizer Subtheories*, Phys. Rev. Lett. **129**, 120403 (2022).

8. The claim that “outcome determinism is not entailed by noncontextuality, and thus should be viewed as an independent assumption in proofs of contextuality” could be strengthened. The relevant point seems to me to be that the assumption isn’t really motivated by anything. If giving up determinism let us recover a simple classical (but stochastic) explanation of the world, quantum foundations as a field would not exist. Physicists simply would accept indeterminism. (There was a debate to this effect in the literature on Bell scenarios, where some folks took the conclusion to simply be that one must reject determinism. But I think those folks lost that debate, and I believe that this is widely accepted by now.)

We discuss the assumption of determinism in Bell’s theorem in more detail in response to the other referee’s remark about the assumption of λ -mediation, around Equation (2).

The referee’s point about indeterminism is a good one, and indeed the “operationalist” approach to Bell’s theorem (as described in [1]) which relies on the 1964 version of the theorem concludes that nature/quantum theory is local and indeterministic and that this settles the question. Therefore, there is a school of physicists which do take the position described by the referee.

For proponents of the 1976 version based on local causality (the “realist” camp to use again the terminology of Calvacanti and Wiseman in [1]), there is no such way out, and as the referee states, they cannot take the conclusion of Bell’s theorem to be a rejection of outcome determinism.

We are not so familiar with the details of the debate referred to by the referee and so cannot comment on which view has won out/is more widely accepted. Since the other referee states that the local realism version of Bell’s theorem is their preferred presentation of it, we have decided to leave the claim as it is, since it applies to both those who view outcome determinism as a well-motivated assumption and those who don’t.

9. The authors do not claim the same sort of device-independence as Bell, but do consider their work a significant step forward towards theory-independent experimental demonstrations. I agree with this, but I think a bit more could be said, or a citation could be given to somewhere that this is discussed more fully, since I think a non-expert reader might not really know how to square these two claims.

Since we did not find existing work explicitly specifying the difference between the two approaches (though there is work such as [2] which makes use of both approaches) we have added a paragraph at the end of Section III. A. explaining some of the main conceptual differences between the approaches.

10. I do not agree with the claim that “It is instructive to be slightly more specific about what the physical system really is... following 43, 44, one can say that we have probed the properties of the effective physical system that is determined by our experimental setup and choice of 100 preparation and measurement procedures”. Strong criticisms have been made against the perspective that a collection of experimental procedures can meaningfully and generally constitute a notion of a system; see for example the discussion in <https://journals.aps.org/prl/abstract/10.1103/PhysRevLett.132.050202>, where it is shown that if one tries to take this point of view on a system, one generally reaches silly conclusions. I am concerned that this (in my opinion problematic) view keeps getting repeated in the literature, without any defence against the criticism made against it. If the authors are nonetheless sympathetic to this perspective, I encourage them to defend the position explicitly.

We agree that this point of view needs much more elaboration, in particular, one that answers the criticism that has been brought forward in the cited work and elsewhere. Since it is simply unnecessary in the present paper, and as such might also confuse the reader, we have removed the statement about the “effective system”.

However, this paragraph also had a second goal: to explain why we are entitled to conclude the certification of generalized contextuality, even if we only implemented 100 of the infinitely many preparations and measurements. This is due to our assumption of tomographic completeness, and due to Selby et al.’s notion of a GPT fragment. To explain this now (without referring to the contentious notion of an effective physical system), we now write:

“It is instructive to be slightly more specific about what the GPT system really is that we have determined. Our experimental choice of 100 preparation and measurement procedures defines an operational theory, to which we have fitted an effective GPT system with state space S and effect space E .” And then it argues as before why the physical GPT is larger, but why we are correct to infer generalized contextuality from this.

Note that we are not claiming any physical relevance for this operational theory; in particular, we are not saying that it defines a physical system. It is just exactly the mathematical

procedure “going from an operational theory to a GPT system” that, when applied to the result of implementing the 100 preparation and measurement procedures, yields the GPT that we plot.

11. Progress was made in answering the question of “what can such experiments tell us if we do not make any assumption of tomographic completeness...” in the authors’ Ref. 57.

We agree that it makes sense to mention this. We have added the following: *“Some progress has recently been made on this question [58], in particular by showing that a notion of relative tomographic completeness is sufficient for demonstrating generalized contextuality, but there are still important open questions on how to apply this to the analysis of concrete experiments.”*

A few more comments are in order. I did not check the experimental details of the work, nor all of the data analysis. However, I did go through the basic methods of the paper, and they are not only sound, but state of the art. Also, I do not believe that the paper introduces any significant new theoretical ideas. However, as I stated above, I think that the experiment will be an inspiration for future experiments that aim to study quantum systems under minimal theoretical assumptions. In addition, the experimental capabilities required for the present experiment seem to me impressive, and perhaps uniquely suited to platforms of this kind. As such, I recommend publication.

We would like to thank the reviewer once again for their thoughtful comments!

II. REVIEW REPORT 2

We are grateful to Reviewer 2 for their thorough assessment of our paper, and for the time and effort they have spent on our work. Even though a few points may have resulted from misunderstandings (for example, Figure 2 is illustrating real measurement results and not simulations), the comments have been extremely helpful for us to improve presentation and clarity. Below, we respond to each comment individually, with the Reviewer’s comments in teal and our answers in black.

Aloy et al analyze data taken on a superconducting circuit experiment from the point of view of generalized contextuality. They claim to have evidence of non-Markovian behaviour and of decoherence without assuming quantum theory.

I find original the idea of measuring a qubit in many different axis to find that a single qubit can exhibit non contextual behaviour. I do find that a discussion about Bell’s 1965 paper on the existence of hidden variable theories for single qubits (where he exposes a 30 year old hidden assumption in a proof in von Neuman’s book of classical impossibility to explain quantum mechanics) is missing. I also find insightful the idea of measuring the rank and the fluctuation of the fit to the matrix to measure the level of non-contextuality of the probabilities measured.

We thank the referee for finding the ideas presented in our study original and interesting. We also thank the referee for pointing out Bell’s 1965 paper. We have added a citation to this paper (now reference [41]) to the end of Subsection II.D (Contextuality and its loss under decoherence). In more detail, we have added the following paragraph:

“It is well-known that the quantum bit is noncontextual in the sense of Kochen and Specker [2], and that it admits of simple hidden-variable models, including one that has already been given by Bell [41]. However, as shown by Spekkens [3], all such models must be contextual according to the generalized notion introduced in Subsection II B. As we have demonstrated above, the same is

true for the “noisy qubits” that describe the superconducting system for sufficiently small evolution times.”

The referee is correct to highlight the similarity between the linearity assumption in von Neumann’s “no hidden variables” proof and the linearity present in the definition of generalized non-contextuality. We stress however that von Neumann concluded that there could not be any hidden-variable model of quantum theory. In our case, we conclude that there cannot be a generalized non-contextual hidden variable model of a qubit. This does not rule out the existence of hidden variables in general, it merely shows that any such model must exhibit generalized preparation contextuality. This is analogous the Bell’s theorem, which does not rule out all hidden-variable models, but only those which are local and realistic.

Note that the linearity assumption that appears in our paper is not postulated, but derived: it follows from the requirement that equivalent preparation procedures are represented by the same distribution over hidden variables (and similarly for measurements), which implies the existence of a linear embedding into the space of classical probability distributions. This requirement is in turn motivated by Leibniz’ Principle of the Identity of the Indiscernibles. We emphasize that this is not our contribution, but part of the well-established definition of generalized contextuality.

While the research direction is interesting, the experimental data is of too low quality and badly presented. This is demonstrated by the fact that figure 2 is simulation and not the data, which is readily available to the authors. I can only assume the data did not look good, and confirm this guess by looking at the data presented. This is uncomfortable since the single qubit experiment here performed could have been done with very high quality using sunlight, a set of polarizing sunglasses and a camera addressing the polarization of light as qubit. Decoherence and non-Markovianity could have been simulated and detected by averaging over different preparations.

Regarding the data acquisition and analysis, we would like to clarify that this is based on many millions of measurement repetitions on a state-of-the-art superconducting circuit experiment. We thus respectfully disagree on the fact that the data we have presented are of low quality. Moreover, we would like to clarify that **Figure 2 is illustrating real measurement results and not simulations**, where every pixel of the 100×100 matrix is averaged over 2000 measurement repetitions. The fact that this figure looks indistinguishable to the eye (also for us) from simulated data is an indication of the high quality and accuracy of our experimental data. We have now clarified in the figure caption that what is plotted are the experimental measurement results.

We disagree that the same experiment could have been performed using sunlight and polarizing sunglasses. Let us first argue informally, and then more formally. Informally, it is important to note that the GPT state vectors (elements of the state space, e.g. of the Bloch ball) describe *probabilities*. The Jones vectors (polarization states) of classical light, however, are best interpreted as *intensities* and not as probabilities. Now consider ensemble E_1 , which describes light that is with 50% probability H-polarized, and with 50% probability V-polarized. Similarly, E_2 is the ensemble of 50% A- and 50% D-polarization, where A and D are for antidiagonal and diagonal, respectively. Now, these ensembles are perfectly distinguishable with a pair of sunglasses! However, in the qubit Bloch ball which describes single photon states, both ensembles would correspond to the same state (the maximally mixed state), and give the same statistics for all measurements. In particular, there are no non-trivial “operational equivalences” (mixtures that yield identical statistics) in the referee’s sunlight example which could be used to establish generalized contextuality.

More formally, consider using the sunglasses to polarize sunlight in the horizontal direction; the state of the light after the polarization is then $|H\rangle^{\otimes n}$ where $n \gg 1$, since there are a very large number of photons n in the sunlight. Similarly for an arbitrary polarization direction one would obtain the state $|\psi\rangle^{\otimes n} = (\alpha|H\rangle + \beta|V\rangle)^{\otimes n}$. As we now show, the properties of a system with these

states are very different from those of a qubit, with pure states $|\psi\rangle = (\alpha|H\rangle + \beta|V\rangle)$.

Observe that the states $|\psi\rangle^{\otimes n}$ are not qubit pure states: they span the symmetric subspace of $(\mathbb{C}^2)^{\otimes n}$ which has dimension $n+1$ which is not \mathbb{C}^2 for $n > 1$.

Alternatively, observe that the overlap between states of different polarizations is given by:

$$|\langle\psi|^{\otimes n}|\phi\rangle^{\otimes n}|^2 = |\langle\psi|\phi\rangle|^{2n} \tag{1}$$

which tends to 0 as $n \rightarrow \infty$. Hence all the polarization states become orthogonal for large n , showing them to be classical (these are also known as SU(2) coherent states). This is in stark contrast to the pure states of a qubit for which the overlap $|\langle\psi|\phi\rangle|^2$ is 0 for orthogonal states only.

Relatedly the mixture $\frac{1}{2}(|H\rangle\langle H|^{\otimes n} + |V\rangle\langle V|^{\otimes n})$ is not equal to the equal mixture of $\frac{1}{2}(|+\rangle\langle +|^{\otimes n} + |-\rangle\langle -|^{\otimes n})$ (where $|\pm\rangle = \frac{1}{\sqrt{2}}(|H\rangle \pm |V\rangle)$), confirming that this is not a qubit, since for the latter an equal mixtures of orthogonal states are indistinguishable (as encoded by the fact they are represented by the same density operator).

The GPT description of such a system would have state space $\text{Conv}(|\psi\rangle\langle\psi|^{\otimes n})$ for all normalized $|\psi\rangle$ in \mathbb{C}^2 . This state space has span corresponding to the Hermitian operators on $\text{Sym}^n(\mathbb{C}^2) \simeq \mathbb{C}^{n+1}$ and hence spans a real vector space of dimension $(n+1)^2$. As $n \rightarrow \infty$, the extremal states $(|\psi\rangle\langle\psi|^{\otimes n})$ have overlap which tends to 0 and hence the state space becomes the space of probability measures over S^2 , i.e. the space of mixed states of a classical system with phase space S^2 .

To observe contextuality in such a system, one would need to prepare states which were not of the form $|\psi\rangle^{\otimes n}$ (these are SU(2) coherent states, which are semi-classical with classical limit $n \rightarrow \infty$), but states which are superpositions of these states, such as Dicke states.

I also disagree with the claim that "Obtaining these conclusions in a theory-independent way establishes that these are properties of the physical system, and not of our theoretical description of the system." This types of claims must be reserved for more elegant experiments that technically exhaust alternative explanation by control experiments. There is not nearly enough evidence here to discard trivial artifacts.

I cannot recommend this paper for publication.

We agree with the general attitude that it is important to be careful on the statements we make, to name your assumptions, and to be aware of potential experimental errors. Still, we have designed the experiment to be analyzable without assuming the validity of quantum theory. In this sense, it probes properties of the system which are independent of our choice of theoretical description. However, we certainly agree that there are *some* assumptions that our analysis makes, and that there is always the possibility of experimental error. We have thus reformulated the sentence (and the subsequent one) as follows:

Since we do not need to assume quantum theory in the analysis of the experiment, these properties are demonstrated to hold irrespective of the theoretical description of the physical system. That is, even if quantum theory were to be overturned by another theory in the future, our conclusions would still hold, as long as our experiment and its analysis have been correctly performed, and our assumption of tomographic completeness (see Methods Subsection I) is satisfied.

We would like to emphasize, however, that our experiment has been very carefully implemented and analyzed. Due to the many repetitions, the quality of the state descriptions is very good; the fact that these states are not close to pure, but quite mixed (84% fidelity) is not detrimental to the conclusions of the experiment, since it is sufficient to prove that a noncontextual hidden-variable model cannot exist for small evolution times. Importantly, these conclusions rely solely on the *statistics* of the experiment, and not on the specific *meaning* or *physical implementation* of the preparation or measurement procedures.

Minor comments:

- Plots are low quality, pixelated and with overlapping legends.

Thank you for bringing this to our attention – we have checked through all figures, and particularly improved the resolution of Figures 3, 6 and 7, which were previously quite pixelated as the reviewer spotted. We also moved the axis titles so that they no longer overlap with tick labels and removed the frame from Figure 3(a).

- I do not feel it is proper to talk about decoherence without assuming quantum theory. Decoherence is only meaningful within quantum theory. Across the paper, the authors do a bad job at drawing the line between quantum ideas and more (or less) general notions regarding probability theory.

We agree that we missed to define decoherence for GPTs, which we have now corrected. A definition can be found in a number of papers, for instance [3] or [4]. We have hence added a new subsection (I, Decoherence in GPTs) to the Methods section that explains this definition and our use of this terminology, and shows that important parts of the phenomenology of decoherence of QT generalize to GPTs.

- The theory agnostic tomography is very inefficient. Quantum tomography of a single qubit takes only three mean values. So while maybe philosophically important it is not practical. So a good reason to do it (or to pay attention to it) needs to be given and explained. There is probably some foundational line of thought that justifies the extra effort and will shine new light on quantum theory, but I cannot think of it and the authors do not provide any.

If we know that we have a qubit of quantum theory, and **if** we are interested in obtaining the description of a single state via tomography, **then** three mean values are sufficient. But here, we are doing something very different and more complicated: we have a physical system about which we assume literally nothing, except that we can prepare and measure it in many different ways (which implies that it can be represented by a GPT system). But

- we do not know whether it is actually a *quantum* system. It could instead be described by an arbitrary convex state space of arbitrary dimension. Our goal is to determine this state space (and to verify that it is quantum **without** assuming it initially).
- Even if we knew / assumed that quantum theory holds, we would not a priori know the Hilbert space dimension d of the system.
- Even if we additionally knew d , we would not know the effective state space. For example, some superselection rule could forbid some superpositions, or decoherence up to time τ could render the system completely or partially classical (indeed we demonstrate that this is roughly what happens for non-zero waiting times τ).

To determine the dimension and the shape of the convex state space, we have to implement lots and lots of state preparations and measurements (and we definitely need much more than three expectation values or probabilities). It turns out that the superconducting platform is ideally suited for this, since the fast repetition rate allows us to gather hundreds of measurements per second, hence our experiment.

We agree with the reviewer that we have not communicated this sufficiently well in early parts of the manuscript. Thus, in the introduction, we have replaced the following sentence

For a quantum bit, the state space is the three-dimensional Bloch ball, and this is reproduced to good accuracy in our experiment.

by the following more informative sentence:

For a quantum bit, the state space is the three-dimensional Bloch ball, but a GPT's state space can be any convex set of any dimension. Without assuming the validity of quantum theory, we determine directly from the data that our superconducting system's normalized state space is likely three-dimensional and in shape close to a ball.

- I am not convinced about the observation of non-Markovianity. Non-Markovianity is not a big deal, many effects can cause that in this experiment. Like resonant coupling with a high-Q mode is a typical happening in these types of experiments. (Why the authors say off-resonant?) In the field of superconducting circuits, spurious two level systems (TLSs) are typically coherent and give non-Markovian effects. They are so common that they impose an important limitation of quantum computations and amount to a large part of the error budget in many-qubit experiments. Now, the authors do not show convincing evidence that they have observed a (mundane) non-Markovian effect. Many Markovian effects can cause an increase of coherence. Imagine a Rabi oscillation under photon loss. When the qubit is close to ground, purity goes up. This may not be what the authors see, but I am just giving it as a trivial example of how increase in purity can be explained by Markovian effects. The point is: the authors have not done nearly enough control experiment to make their grand claims. In this particular case, the author could do control experiments, find that mode or TLS and confirm that what they discovered via a theory agnostic experiment is in fact there. That would be cute, but the claim by itself is badly supported by the analysis and the data.

We agree with the referee with the fact that observing non-Markovian dynamics is, per se, not a surprise. Please also see the related comment by the other reviewer, and our answer to it. In a nutshell, we have added a paragraph at the end of Section II, explaining why contextuality and non-Markovianity are very different phenomena here. In particular, non-Markovianity is indeed not a big deal in itself, and there are many reasons why it may appear, even in fully classical systems.

However, what we *do* consider a valuable achievement is the *theory-independent verification* of non-Markovianity in the system, which is the goal of our analysis. In contrast, we do **not** aim at characterizing or explaining *where* this non-Markovianity is coming from. In order to achieve this theory-independent verification, we prove rigorously that Markovian time evolutions cannot increase the volume of the state space for any GPT, quantum or not (see in particular our improved explanation in Subsection II.E, answering also a request by the other reviewer). Hence, seeing the volume increase proves non-Markovianity, without assuming quantum theory. In particular, we are **not** arguing with an “increase in purity” as the referee seems to suggest, and which we agree could be explained by Markovian effects.

As correctly pointed out, non-Markovianity in experiments such as the one considered here can often be attributed to coupling with high-Q modes and two-level-systems (i.e. impurities and defects in the qubit's solid-state environment). In our manuscript, we have mentioned off-resonant coupling because in general this can also happen, but we have now decided to remove this term since we do not know whether it is indeed this sort of coupling, and not a different one, which leads to the observed non-Markovianity. Let us again emphasize that clarifying the source of this non-Markovianity has **not** been a goal of our analysis, in which we have deliberately restricted ourselves to what can be said without assuming the validity of quantum theory.

- "dispersive-readout" was not "mentioned above". They called it "measure by dispersive coupling". No need to be inconsistent with terminology.

Both the words we have used are common terminology in the superconducting circuit, but also in the cavity electrodynamics, community [5]. In the beginning of the experimental paragraph we explain that the qubit measurement is enabled by operating in the (strong) dispersive coupling regime of the Jaynes-Cummings Hamiltonian, which is what is called in short "dispersive read-out". To improve clarity of our text, we have now written: "Finally, the qubit's state is measured in the computational z -basis $\{|0\rangle, |1\rangle\}$ by the standard circuit-QED read-out technique in the strong-dispersive-coupling regime, as mentioned above."

- I may be wrong here, but I don't think so: equation (1) is poorly introduced. It assumes a factorization that is not justified by probability theory. I understand it is equivalent to $P(A, B|C) = P(A|C)P(B|C)$. I think I know why the authors write this, it is sadly usually found in this context and known as the Reichenbach factorization principle of common cause (and used by Bell in his original proof, improved by CHSH, in page 116 of Nielsen and Chuang there is a derivation without this factorization, which I deem to be the fundamentally correct one). However, the fundamentally correct factorization is (Bayes) $P(A, B|C) = P(A|B, C)P(B|C)$. They are only equivalent, I understand, if the common cause C (λ in their notations) makes any of these probabilities be $P = P^2 = 0, 1$. i.e. if C makes the system completely deterministic.

The referee is correct that Equation (1) requires a factorization assumption, which we did not explicitly state. Before discussing this, let us emphasize that Equation (1) is **not a new assumption of our publication**, but uniformly accepted by the community of researchers working on generalized contextuality as part of the definition. In our paper, we are simply *using* this notion of contextuality, and solving the task of detecting it (and its time-dependence) in a theory-independent way in a superconducting system. For this reason, we think it is justified that we are not discussing the derivation of Equation (1) in detail in our paper, but are instead citing the relevant literature for justification. We have added reference [6] (now reference [4] in the paper) when introducing generalized contextuality which explicitly discusses the assumption of λ -mediation which is needed for the factorisation assumption. We have now also explicitly mentioned λ -mediation when contrasting our methods to the device independent framework in Section III. A.

We agree that the referee's question is certainly thoughtful and interesting in general. The aforementioned factorization requires the assumption of λ -mediation, and is indeed not always properly stated in presentations of ontological models. The justification for this assumption is that the hidden variables should encode all the properties of the system, namely that the probability of measurement outcomes should depend solely on the ontic state. As argued in [6], this is logically distinct from a non-contextuality assumption, and is rather an assumption about the causal structure of the experiment, which manifests as a 'screening off' of the preparation by the ontic state.

Explicitly λ -mediation is the assumption that $p(O|M, P, \lambda) = p(O|M, \lambda)$.

As emphasised in [6], a further assumption is also made:

$$p(\lambda|M, P) = p(\lambda|P) \tag{2}$$

which is called measurement independence, and is justified by no retro-causality.

Using standard probability theory (Bayes rule), we have:

$$p(O|M, P) = \sum_{\lambda} P(O, \lambda|M, P) = \sum_{\lambda} P(O|M, P, \lambda)P(\lambda|M, P). \quad (3)$$

If we furthermore assume λ -mediation and measurement independence, we have:

$$p(O|M, P) = \sum_{\lambda} P(O|M, \lambda)P(\lambda|P) \quad (4)$$

For a criticism of the assumption of λ -mediation, see for example [7].

For a deterministic model it is indeed the case as the referee points out that λ -mediation is trivially satisfied, i.e. $P(O|M, P, \lambda) = p(O|M, \lambda)$ since $P(O|M, P, \lambda) = 0, 1$.

Let us briefly provide a short digression on Bell which will show an analogy with this case here. Bell's theorem is typically proven from two distinct sets of starting assumptions:

- locality + realism (i.e. determinism) from Bell's 1964 version of the theorem (which the one presented on p.116 of Nielsen and Chuang cited by the referee);
- local causality from Bell's 1976 version of the theorem.

The combination of locality + determinism together imply Bell's theorem. However one can appeal instead to local causality to derive Bell's theorem. Here outcome determinism is not assumed, but rather screening off properties of different space-time regions are used to derive factorisability.

Here we see the analogy with the derivation of the equation $p(O|M, P, \lambda) = p(O|M, \lambda)$ which can be motivated by either an appeal to determinism or by appealing to λ -mediation, which is a screening off condition, too. This is analogous to the derivation of Bell inequalities from either locality + determinism or local causality.

-
- [1] H. M. Wiseman and E. G. Cavalcanti, *Causarum Investigatio and the Two Bell's Theorems of John Bell*, arXiv:1503.06413 (2015). <https://arxiv.org/abs/1503.06413>
 - [2] A. B. Sainz, Y. Guryanova, A. Acín, and M. Navascués, *Almost-Quantum Correlations Violate the No-Restriction Hypothesis*, Phys. Rev. Lett. **120**, 200402 (2018).
 - [3] J. G. Richens, J. H. Selby, and S. W. Al-Safi, *Entanglement Is Necessary for Emergent Classicality in All Physical Theories*, Phys. Rev. Lett. **119**, 080503 (2017). <https://journals.aps.org/prl/abstract/10.1103/PhysRevLett.119.080503>
 - [4] M. P. Müller and A. J. P. Garner, *Testing Quantum Theory by Generalizing Noncontextuality*, Phys. Rev. X **13**, 041001 (2023). <https://journals.aps.org/prx/abstract/10.1103/PhysRevX.13.041001>
 - [5] A. Blais, A. L. Grimsmo, S. M. Girvin, and A. Wallraff, *Circuit quantum electrodynamics*, Rev. Mod. Phys. **93**, 025005 (2021).
 - [6] L. Catani and M. Leifer, *A mathematical framework for operational fine tunings*, Quantum **7**, 948 (2023).
 - [7] E. Adlam, *Spooky Action at a Temporal Distance*, Entropy **20**(1), 41 (2018). <https://www.mdpi.com/1099-4300/20/1/41>